# BiHDTrans: binary hyperdimensional transformer for efficient multivariate time series classification

## Abstract

The proliferation of Internet-of-Things (IoT) devices has led to an unprecedented volume of multivariate time series (MTS) data, requiring efficient and accurate processing for timely decision-making in resource-constrained edge environments. Hyperdimensional (HD) computing, with its inherent efficiency and parallelizability, has shown promise in classification tasks but struggles to capture complex temporal patterns, while Transformers excel at sequence modeling but incur high computational and memory overhead. We introduce BiHDTrans, an efficient neurosymbolic binary hyperdimensional Transformer that integrates self-attention into the HD computing paradigm, unifying the representational efficiency of HD computing with the temporal modeling power of Transformers. Empirically, BiHDTrans outperforms state-of-the-art (SOTA) HD computing models by at least 14.47% and achieves 6.67% higher accuracy on average than SOTA binary Transformers. With hardware acceleration on FPGA, our pipelined implementation leverages the independent and identically distributed properties of high-dimensional representations, delivering 39.4× lower inference latency than SOTA binary Transformers. Theoretical analysis shows that binarizing in holographic high-dimensional space incurs significantly less information distortion than directly binarizing neural networks, explaining BiHDTrans's superior accuracy. Furthermore, dimensionality experiments confirm that BiHDTrans remains competitive even with a 64% reduction in hyperspace dimensionality, surpassing SOTA binary Transformers by 1–2% in accuracy with 4.4× less model size, as well as further reducing the latency by 49.8% compare to the full-dimensional baseline. Together, these contributions bridge the gap between the expressiveness of Transformers and the efficiency of HD computing, enabling accurate, scalable, and low-latency MTS classification.

## 1 Introduction

The rapid proliferation of Internet-of-Things (IoT) devices has led to an unprecedented volume of sensor data being generated at the network edge (Ahmed et al., 2017). A large portion of this data takes the form of multivariate time series (MTS), such as physiological signals in wearable healthcare devices, vibration profiles in industrial monitoring, and environmental measurements in smart cities (Zeng et al., 2023). Efficient and accurate processing of MTS data is therefore critical for enabling timely decision-making in latency-sensitive and resource-constrained environments.

Hyperdimensional (HD) computing has recently emerged as a brain-inspired learning paradigm that represents and manipulates information in high-dimensional spaces using holographic, distributed encodings (Kanerva, 2009). HD computing is inherently efficient, parallelizable, and compatible with low-precision operations, making it a natural candidate for edge inference. Prior work has demonstrated the effectiveness of HD computing across various classification tasks, including language recognition, bio-signal analysis, and sensor-based activity recognition (Ge & Parhi, 2020). However, existing HD temporal encoding methods for time series rely on simple schemes such as position binding or sliding windows (Rahimi et al., 2019), which fail to capture long-range depen-

dencies and complex temporal patterns. In contrast, Transformers excel at modeling such dependencies through self-attention (Vaswani et al., 2017), and have recently achieved state-of-the-art (SOTA) performance in MTS classification (Eldele et al., 2024). However, their accuracy gains come at the expense of heavy computation and memory footprints, which severely limit their deployment on edge devices and embedded hardware.

In this work, we present BiHDTrans[1], an efficient neurosymbolic **Bi**nary **H**yper**D**imensional **Trans**former framework that unifies the representational efficiency of HD computing with the temporal modeling capabilities of Transformers. By integrating self-attention into the HD computing paradigm, BiHDTrans effectively captures both intra-variable dynamics and cross-variable correlations in MTS. Empirically, it achieves at least 14.47% higher classification accuracy than SOTA HD computing methods and 6.67% higher accuracy than SOTA binary Transformers. To demonstrate practical efficiency, we design a pipelined FPGA implementation that leverages the independent and identically distributed (i.i.d.) properties of high-dimensional representations, enabling a fully parallelized flow that eliminates sequential bottlenecks and achieves 39.4× lower inference latency than SOTA binary Transformers on a Xilinx Artix-7 device. We further provide theoretical proofs showing that binarizing a Transformer in holographic high-dimensional space incurs significantly less information distortion than directly binarizing neural network weights and activations, thereby justifying the superior accuracy of BiHDTrans. Finally, by exploring dimensionality tradeoffs, we show that BiHDTrans remains competitive even with a 64% reduction in hyperspace dimensionality, surpassing SOTA binary Transformers by 1–2% in accuracy with 4.4× less model size,as well as further reducing the latency by 49.8% compare to the full-dimensional baseline, offering a tunable trade-off between efficiency and performance.

## 2 RELATED WORK

### 2.1 HYPERDIMENSIONAL COMPUTING

HD computing, also known as vector symbolic architecture (VSA), comprehensively introduced by Kanerva (2009), is a brain-inspired paradigm that represents information as randomly generated high-dimensional vectors (i.e., hypervectors), enabling robust, holographic, and highly parallel computation. Its potential was first demonstrated by Rahimi et al. (2016) through EMG-based hand gesture recognition, which showcased the framework's efficiency in biosignal processing. In recent years, numerous extensions have been proposed to enhance accuracy, adaptability, and hardware efficiency. SemiHD (Imani et al., 2019) incorporates semi-supervised learning into the HD framework, while MulTa-HDC (Chang et al., 2021) extends HD computing to multi-task learning. Processing-in-Memory (PIM) architecture designed such as HyDREA (Morris et al., 2021) further improve robustness and efficiency. NeuralHD (Zou et al., 2021) introduces dynamic encoders to enhances adaptability on edge devices, whereas DistHD (Wang et al., 2023) and RefineHD (Vergés et al., 2023) further advance the field by introducing learner-aware dynamics and enabling single-pass adaptive learning, respectively. Furthermore, Store-n-Learn (Gupta et al., 2022) explores in-storage classification and clustering across flash hierarchies. LeHDC (Duan et al., 2022a) enhances representational precision by introducing learning-based classifier, and Yu et al. (2024) later proposed a fully learnable HD framework with an ultratiny accelerator for edge-side applications, demonstrating the feasibility of compact deployment. Referred to the reviews by Ge & Parhi (2020) and Vergés et al. (2025) for broader overviews.

### 2.2 TIME SERIES FOUNDATION MODELS

Recent advances have led to the emergence of time series foundation models (TSFMs), which pretrain large Transformer-based architectures to enable time series representation learning (Liang et al., 2024). Representative examples include MOMENT (Goswami et al., 2024), a multi-task TSFM pretrained on a large "time-series pile"; Mantis (Feofanov et al., 2025), a ViT-based model for universal time-series classification; and NuTime (Lin et al., 2024), which improves general-purpose temporal representation learning through large-scale pretraining. Other recent TSFM variants, such as Kairos citepfeng2025kairos and TimeDiT (Cao et al., 2024), further explore adaptive tokenization and unified probabilistic modeling. These models demonstrate strong representational power

---

[1]https://anonymous.4open.science/r/BiHDTrans-4E55

and notable generalization across domains. However, like earlier Transformer-based approaches for time-series modeling (Wu et al., 2021; Zhou et al., 2022; Kitaev et al., 2020; Zhang & Yan, 2023), TSFMs also rely on full-precision networks, resulting in high computational and memory costs. This makes their deployment on resource-constrained edge hardware challenging.

## 3 PRELIMINARIES

### 3.1 HD OPERATIONS

HD computing represents information as high-dimensional vectors (hypervectors) in hyperspace, whose dimensionality typically ranges from thousands to tens of thousands (e.g. $D = 10000$). The following i.i.d. operations are defined in hyperspace to manipulate and combine hypervectors (Ge & Parhi, 2020). **Bundling** ($\odot$)**:** element-wise addition of hypervectors, producing a representation that preserves similarity to all inputs. **Binding** ($\oplus$)**:** element-wise multiplication (Hadamard product), producing a representation that is dissimilar to each input, often used to associate or "bind" two entities. **Cyclic permutation** ($\rho^k$)**:** shifting the coordinates of a hypervector by $k$-bits, commonly used to represent order or positional information.

In hyperspace, cosine similarity is used to measure the similarity between two hypervectors: $\cos(\mathcal{H}_1, \mathcal{H}_2) = \frac{\mathcal{H}_1^T \mathcal{H}_2}{\|\mathcal{H}_1\|\|\mathcal{H}_2\|}$. For binary hypervectors, this is equivalent to the more efficient Hamming distance: $\text{Ham}(\mathcal{H}_1, \mathcal{H}_2) = \frac{|\mathcal{H}_1 \neq \mathcal{H}_2|}{D}$, as $\mathcal{H}_1^T \mathcal{H}_2 = |\mathcal{H}_1 = \mathcal{H}_2| - |\mathcal{H}_1 \neq \mathcal{H}_2|$, $\|\mathcal{H}_1\|\|\mathcal{H}_2\| = D$ and $|\mathcal{H}_1 = \mathcal{H}_2| + |\mathcal{H}_1 \neq \mathcal{H}_2| = D$, which results in $\cos(\mathcal{H}_1, \mathcal{H}_2) = 1 - 2 \cdot \text{Ham}(\mathcal{H}_1, \mathcal{H}_2)$.

### 3.2 HD MAPPING AND ENCODING

#### 3.2.1 MAPPING TO HYPERSPACE

Consider a sample $\boldsymbol{F} = \{f_1, f_2, \ldots, f_N\}$, where $f_i$ denotes the value of $i$-th feature. Feature positions and feature values are independently mapped into hyperspace with randomly generated binary hypervectors. The position hypervectors ($\mathcal{F}$) are orthogonal to maintain the independent and discrete representation of the feature positions: $\text{Ham}(\mathcal{F}_i, \mathcal{F}_j) \approx 0.5$ for $i \neq j$. Alternatively, the feature values are quantized and mapped into correlated value hypervectors ($\mathcal{V}$) such that their pairwise Hamming distances reflect the correlation of quantized values: $\text{Ham}(\mathcal{V}_i, \mathcal{V}_j) \propto \frac{f_i - f_j}{max - min}$, where $f_i, f_j \in [min, max]$ are two samples in the value range.

#### 3.2.2 SPATIAL AND TEMPORAL HD ENCODING

Spatial encoding produces a single binary hypervector representation for one multivariate sample. This is done via the hash-table encoding, by binding the position hypervector $\mathcal{F}_i$ and value hypervector $\mathcal{V}_i$ for each feature $i$, and followed by binarization: $\mathcal{S} = \text{sign}\left(\sum_{i=1}^{N} \mathcal{F}_i \odot \mathcal{V}_i\right)$, where $S \in \{-1, +1\}^D$.

For MTS, after mapping and encoding the multivariate sample at each timestamp, a sequence of hypervectors is produced at each time step: $\{\mathcal{S}_1, \mathcal{S}_2, \ldots, \mathcal{S}_L\}$, where $L$ is the sequence length. These temporal hypervectors are further combined into a single representation via cyclic permutation and binding: $\mathcal{T} = \prod_{t=1}^{L} \rho^t(\mathcal{S}_t)$, where $\mathcal{S}_t$ is the sample at the $t$-th timestamp.

### 3.3 HD CLASSIFICATION

**Training**. For vanilla HD classifier (Heddes et al., 2023), each class is represented by the centroid hypervector (prototype) of its spatial-temporal encoded training samples. For class $k$, the prototype is computed as: $\mathcal{C}_k = \text{sign}\left(\sum_{\mathcal{T} \in \Omega_k} \mathcal{T}\right)$, where $\Omega_k$ denotes the set of training hypervectors belonging to class $k$, and $\mathcal{C}_k \in \{-1, +1\}^D$ is stored in the associative memory (AM). **Inference**. For a query hypervector $\mathcal{T}_q$, similarity is computed between $\mathcal{T}_q$ and each class prototype in AM. The predicted label corresponds to the prototype with highest similarity: $\hat{y} = \arg\max_k \text{Ham}(\mathcal{T}_q, \mathcal{C}_k)$.

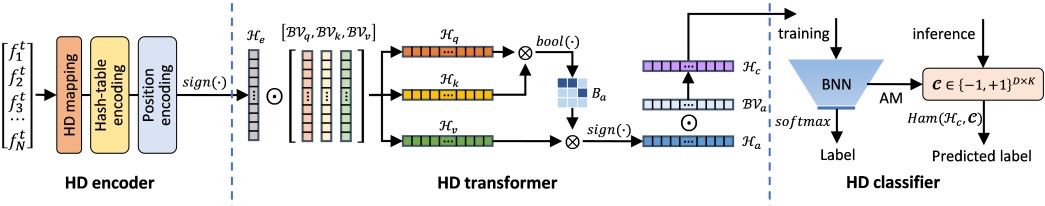

Figure 1: The overall framework of BiHDTrans.

## 4 THE RISE OF BIHDTRANS

Figure 1 illustrates the overall architecture of BiHDTrans, which consists of three main modules: the HD encoder, the HD Transformer, and the HD classifier. The HD encoder maps multivariate time series at each time step into a binary hypervector, enabling efficient binary and i.i.d. HD operations. The HD Transformer then processes these binary hypervectors, performing self-attention entirely within the HD computing domain to capture temporal dependencies across variables. Finally, the HD classifier takes the Transformer's output and trains the AM as an equivalent binarized neural network (BNN) layer. During inference, labels are predicted via associative memory search using Hamming distance, as described in Section 3.3.

### 4.1 HD ENCODER

The HD encoder is responsible for converting each multivariate observation in the time series into a binary high-dimensional representation suitable for subsequent HD-domain self-attention. The mapping of feature positions and values into hyperspace follows the process described in Section 3.2.1.. After hyperspace mapping, hash-table encoding is applied by binding each position hypervector $\mathcal{F}_i$ with its corresponding value hypervector $\mathcal{V}_i^t$ at time step $t$, followed by bundling across all features. Positional encoding is then incorporated via a cyclic permutation $\rho^t(\cdot)$ to embed temporal order, and the resulting vector is binarized using the element-wise sign function. Formally, the encoded hypervector at token (i.e., time step) $t$ is given by

$$\mathcal{H}_e^t = \text{sign}\left(\rho^t\left(\sum_{i=1}^N \mathcal{F}_i \odot \mathcal{V}_i^t\right)\right), \tag{1}$$

where $\mathcal{H}_e^t \in \{-1, +1\}^D$. Collectively, the encoder produces the fully binarized sequence $\mathcal{H}_e = [\mathcal{H}_e^1, \mathcal{H}_e^2, ..., \mathcal{H}_e^L]$ which serves as the input to the HD Transformer.

**Theorem 1** *Let $X \sim \mathcal{N}(0, \sigma)$, and the dimension of the hyperspace $D \to \infty$. Given that the quantization level $q \geq 3$ for the real-valued feature, the information distortion from binarizing $X$ in hyperspace is lower than that incurred by directly binarizing the real-valued data.*

Theorem 1 establishes the theoretical foundation for BiHDTrans, demonstrating that performing fully binarized self-attention in the HD computing paradigm offers potential advantages over purely NN-based approaches that use fully-binarized weights and activations in Transformer models for MTS classification. By operating entirely within the binary high-dimensional space, the HD encoder ensures both representational robustness and hardware-friendly efficiency. The full proof of Theorem 1 can be found in Appendix B.3, while the empirical validation is provided in Appendix C.1.

### 4.2 HD TRANSFORMER

To implement self-attention entirely within the binarized HD computing paradigm, we replace the fully connected (FC) linear layers in conventional Transformers with a binding operation between the input hypervector and a trainable binary binding hypervector BV. This substitution is valid because binding in HD computing—element-wise multiplication between two same-sized hypervectors—can be expressed as multiplication by a diagonal weight matrix in a linear layer (Duan et al.,

2022b). Thus, for every attention head, the queries $\mathcal{H}_q$, keys $\mathcal{H}_k$, and values $\mathcal{H}_v$ hypervector for self-attention can be derived as:

$$\mathcal{H}_q = \mathcal{H}_e \odot \mathcal{BV}_q, \quad \mathcal{H}_k = \mathcal{H}_e \odot \mathcal{BV}_k, \quad \mathcal{H}_v = \mathcal{H}_e \odot \mathcal{BV}_v, \tag{2}$$

where $\mathcal{BV}_q, \mathcal{BV}_k, \mathcal{BV}_v \in \{-1, +1\}^D$ are trainable. The binarized attention score matrix is then computed as:

$$B_a = \text{bool}(\mathcal{H}_q \cdot \mathcal{H}_k), \quad B_a \in \{0, 1\}^{L \times L}, \tag{3}$$

where the dot product is performed in integer arithmetic and bool function maps positive values to 1 and non-positive values to 0. Unlike conventional attention, the attention scores are binarized directly and produces discrete selection masks, allowing for omission for both nonlinear dimension normalization $(1/\sqrt{d})$ and nonlinear activation (e.g., softmax).

Subsequently, the HD-based self-attention output is given by matrix multiplication between $B_a$ and $\mathcal{H}_v$. In HD computing terms, this corresponds to selective bundling, where among $\mathcal{H}_v^t$ across all tokens. The selection for the output of token $t$ is determined by the $t$-th row of $B_a$:

$$\mathcal{H}_a^t = \text{sign}\left(\sum_{i=1}^{L} \mathcal{H}_v^i \cdot b_{t,i}\right), \tag{4}$$

where $b_{t,i} \in \{0, 1\}$ is the element at the $t$-th row and $i$-th column of $B_a$ and $\mathcal{H}_a^t \in \mathcal{H}_a = [\mathcal{H}_a^1, \mathcal{H}_a^2, ..., \mathcal{H}_a^L]$. Finally, the aggregated representation is bound with a trainable binary hypervector $\mathcal{BV}_a$ to produce the head output:

$$\mathcal{H}_c = \mathcal{H}_a \odot \mathcal{BV}_a. \tag{5}$$

**Theorem 2** *Let $\{V_i\}_{i=1}^N \in \{-1, +1\}^D$ be a set of binary hypervectors, and let $w = (w_1, w_2, \ldots, w_N) \in \mathbb{R}^N$ be a set of real-valued weights. The weighted sum with real-valued and binarized weights are given by: $Y = \text{sign}\left(\sum_{i=1}^N w_i \cdot V_i\right)$ and $Y' = \text{sign}\left(\sum_{i=1}^N w_{qi} \cdot V_i\right)$, where $w_{qi} \in 0, 1$ is the binarized weight (mask). For any fixed $\alpha \in (0, \frac{1}{2})$, there exists a constant $C(w, \alpha) > 0$ such that as $D \to \infty$, the information distortion between $Y$ and $Y'$ is bounded by $C(w, \alpha)$ and converges to zero with high probability.*

Theorem 2 focuses on the binarization of weights for the bundling operation rather than the binarization of the hypervectors themselves. This makes Theorem 2 an essential complement to Theorem 1, supporting the idea that the information distortion due to binarizing the attention scores in the HD Transformer is more controllable and predictable compared to binarizing attention scores in the real-valued NN domain. The full proof of Theorem 2 can be found in Appendix B.4, while the empirical validation is provided in Appendix C.2.

Unlike conventional Transformers, our HD Transformer omits the feed-forward (FF) block after self-attention. In standard architectures, the FF block serves to re-project and nonlinearly transform the output for richer feature interaction across dimensions (Kobayashi et al., 2024). In our case, the mapping into holographic high-dimensional space already distributes information across all dimensions in an i.i.d. manner. Thus, each dimension is equally informative, and further projection by an FF block theoretically provides no additional expressiveness while only adding unnecessary cost. Therefore, $\mathcal{H}_c$ can be directly passed to the classification stage.

## 4.3 HD CLASSIFIER

The vanilla HD classifier described in Section 3.3 is highly efficient, as it trains the associative memory (AM) via direct superposition. However, it typically yields sub-optimal accuracy on complex classification tasks (Imani et al., 2020b). Moreover, in our case, the vanilla approach can only train the AM but cannot optimize the binding hypervector BVs used in the HD Transformer. To address this, we employ a learning-based HD computing classifier (LeHDC) (Duan et al., 2022a), which treats the AM prototypes $\mathcal{C} \in \{-1, +1\}^{D \times K}$ as the weights of an equivalent BNN. This formulation allows both $\mathcal{C}$ and $\mathcal{BV}$s to be learned jointly via backpropagation.

During training, we maintain dense real-valued parameters $\mathcal{C}_d$ and $\mathcal{BV}_d^{(j)}$ for training, which are binarized as $\mathcal{C} = \text{sign}(\mathcal{C}_d)$ and $\mathcal{BV}^{(j)} = \text{sign}(\mathcal{BV}_d^{(j)})$ during forward propagation. In backpropagation, the straight-through estimator (STE) (Bengio et al., 2013) is used to pass gradients through

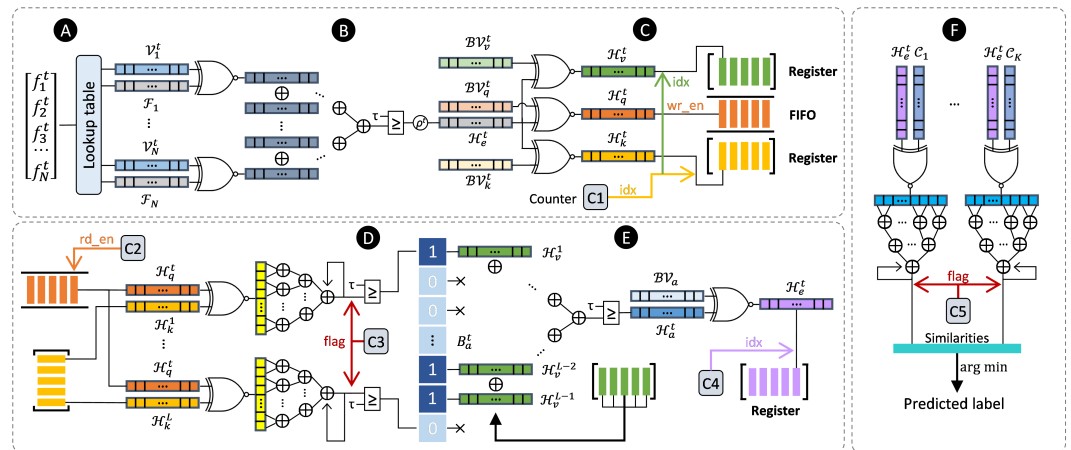

Figure 2: FPGA implementation of BiHDTrans.

the sign function. Specifically, we simplify the output of BiHDTrans as $\mathcal{O} = \text{sign}(\mathcal{C}_d) \cdot (\mathcal{H}^{(j)} \odot \text{sign}(\mathcal{BV}_d^{(j)}))$ for clearer illustration. With the cross-entropy loss $L$, the updates are:

$$\frac{\partial L}{\partial \mathcal{C}_d} = \frac{\partial L}{\partial \mathcal{O}} \cdot \frac{\partial \mathcal{O}}{\partial \text{sign}(\mathcal{C}_d)} = \frac{\partial L}{\partial \mathcal{O}} \cdot \mathcal{H}^{(j)} \odot \text{sign}(\mathcal{BV}_d^{(j)}),  \quad (6)$$

$$\frac{\partial L}{\partial \mathcal{BV}_d^{(j)}} = \frac{\partial L}{\partial \mathcal{O}} \cdot \frac{\partial \mathcal{O}}{\partial \text{sign}(\mathcal{BV}_d^{(j)})} = \frac{\partial L}{\partial \mathcal{O}} \cdot \text{sign}(\mathcal{C}_d) \cdot \mathcal{H}^{(j)},  \quad (7)$$

with $\mathcal{C}_d$ and $\mathcal{BV}_d^{(j)}$ clipped to $[-1, +1]$.

While this LeHDC-based training introduces more computational overhead compared to the vanilla classifier, the inference process remains identical to the efficient method in Section 3.3—relying only on binary hypervectors and associative matching. As will be detailed in Section 5, our pipelined FPGA design accelerates inference process by exploiting the i.i.d. properties of BiHDTrans.

## 5 HARDWARE ACCELERATION OF INFERENCE

Field-programmable gate arrays (FPGAs) are well-suited for accelerating hyperdimensional (HD) computing due to their ability to handle large volumes of simple bitwise operations with high efficiency (Imani et al., 2021). However, edge-class FPGA devices typically lack the resources to process all D dimensions of a hypervector in a single pass. To address this, we adopt a pipelined dataflow design that processes d dimensions in parallel across cascaded inputs, where $d < D$. For BiHDTrans, this constraint is further tightened to $d < D/N_h$, where $N_h$ is the number of attention heads. Figure 2 illustrates the complete FPGA-accelerated inference pipeline of BiHDTrans. To improve logic efficiency, bipolar hypervectors $\{-1, +1\}^D$ are represented in binary form $\{0, 1\}^D$, and bitwise multiplication is implemented with XNOR gates.

The process begins with feature mapping: real-valued multivariate time series inputs are converted into value and position hypervectors via a pre-defined lookup table (Ⓐ). Given the streaming nature of time series and limited hardware resources, samples are processed sequentially. For each time step, the position and value hypervectors are bound in parallel using XNOR, and the results are bundled through a tree-structured adder, binarized, and cyclically permuted to form the encoded hypervector $\mathcal{H}_e^t$ (Ⓑ). Next, $\mathcal{H}_e^t$ is bound with pre-trained binding hypervectors to generate the query, key, and value hypervectors ($\mathcal{H}_q^t$, $\mathcal{H}_k^t$, $\mathcal{H}_v^t$). The key and value hypervectors are stored in indexed registers, using indexes provided by counter C1, while the query hypervectors are streamed into a FIFO for single-pass access (Ⓒ).

Once all-time steps are encoded, the attention computation begins. Queries are sequentially read from the FIFO under the control of counter C2. Each $\mathcal{H}_q^t$ is bound with all stored keys $\mathcal{H}_k^i$ ($i = 1, ..., L$), and attention scores are obtained through parallel tree adders followed by binarization (Ⓓ).

Table 1: Comparison to SOTA HD computing and SOTA binary Transformers

| | Japanese-Vowels | Heartbeat | Spoken-Arabic-Digits | Face-Detection | PEMS-SF | Racket-Sports | Epilepsy | Average |
|---|---|---|---|---|---|---|---|---|
| Full Precision | *98.82* | *78.05* | *96.73* | *63.05* | *85.55* | *84.87* | *96.38* | *86.22* |
| BiBERT | 96.49 | **79.51** | 86.68 | 54.40 | 76.30 | 71.71 | 73.91 | 77.00 |
| BiT | 95.68 | 75.61 | 87.13 | 54.31 | 78.61 | 75.66 | 76.81 | 77.69 |
| BiViT | 95.95 | 76.59 | 73.26 | 54.48 | 75.14 | 62.50 | 64.49 | 71.77 |
| Vanilla HDC | 18.65 | 52.68 | 10.23 | 49.21 | 49.71 | 28.95 | 22.46 | 33.13 |
| QuantHD | 21.08 | 55.12 | 10.23 | 51.22 | 55.49 | 30.92 | 31.88 | 36.56 |
| LeHDC | 38.38 | 54.15 | 11.64 | 51.62 | 58.38 | 60.53 | 30.43 | 43.59 |
| DistHD | 94.86 | 72.68 | 95.09 | 58.71 | 47.40 | 69.74 | 50.72 | 69.89 |
| BiHDTrans | **97.30** | 77.56 | **96.41** | **61.18** | **87.28** | **83.82** | **86.96** | **84.36** |

Since only d dimensions can be processed at once, partial sums are accumulated and controlled by counter C3, which validates outputs after $D/N_h$ dimensions. The resulting binary attention mask $B_a^t$ is then applied to the value hypervectors, producing the attention-weighted hypervector $\mathcal{H}_a^t$. After bundling, binarization, and binding, the t-th token representation $\mathcal{H}_v^t$ is produced and stored in a register with index provided by counter C4 (**E**).

For classification, each token hypervector $\mathcal{H}_v^t$ is bound with all prototype class hypervectors $\mathcal{C}_k$. The resulting similarities are computed using parallel tree adders and accumulated under counter C5, which validates outputs after all $D$ dimensions are processed. The final prediction is obtained by sorting the similarities (**F**). This hardware design exploits both fine-grained parallelism at the bitwise level and coarse-grained pipelining across tokens, enabling resource-constrained FPGAs to efficiently perform BiHDTrans inference with minimal latency.

## 6 EXPERIMENTS AND RESULTS

We evaluate BiHDTrans by comparing its performance with SOTA models across seven well-known MTS datasets from the latest archive (Ruiz et al., 2021), representing common IoT applications including physiological monitoring, speech recognition, GPS tracking, and IMU-based activity recognition. These datasets vary in feature size and time window length, covering a broad range of real-world IoT data.

BiHDTrans is compared against both SOTA HD computing models (QuantHD (Imani et al., 2020a), LeHDC (Duan et al., 2022a), DistHD (Wang et al., 2023) and SOTA fully-binarized Transformers (BiBERT (Qin et al., 2022), BiT (Liu et al., 2022), BiViT (He et al., 2023)). The models are implemented in PyTorch and/or TorchHD, trained and tested on an NVIDIA GeForce RTX 3090 GPU, and deployed on a Xilinx Artix-7 xc7a200tfbg484-2 FPGA for runtime latency evaluation. Both BiHDTrans and binary Transformers use a single Transformer encoder block and output only the final token for MTS classification, ensuring a fair comparison and efficiency for edge deployment. While these binary Transformers are designed for more complex tasks like computer vision and LLMs, accompanied by tailored training techniques such as multi-step binarization and knowledge distillation (Xiao et al., 2025), BiHDTrans and SOTA models in our experiments exclude these techniques for a fairer evaluation.

### 6.1 COMPARISON WITH SOTA MODELS

Table 1 shows that BiHDTrans outperforms SOTA HD computing models by at least 14.47% in classification accuracy across the evaluated MTS datasets. While these HD models enhance the vanilla HD mapper and classifier, they fail to improve the temporal encoder, limiting performance. Compared to SOTA binary Transformer models, BiHDTrans achieves 6.67% higher accuracy on average, validating the theorems which claim that BiHDTrans incurs less information distortion during binarization, resulting in optimal accuracy with a fully binarized pipeline. The Full Precision Transformer is also included as a reference upper bound under the same simplified architecture (single block, last-token output).

Table 2: Inference latency on Artix-7 FPGA (unit: μs).

| | Japanese-Vowels | Heartbeat | Spoken-Arabic-Digits | Face-Detection | PEMS-SF | Racket-Sports | Epilepsy | Average |
|---|---|---|---|---|---|---|---|---|
| BiBERT | 45.35 | 527.37 | 127.35 | 187.56 | 2782.66 | 46.31 | 272.84 | 569.92 |
| BiT | 46.68 | 536.22 | 129.96 | 189.65 | 2786.29 | 47.66 | 277.73 | 573.46 |
| BiViT | 47.61 | 548.43 | 132.81 | 191.57 | 2790.67 | 48.62 | 284.00 | 577.67 |
| BiHDTrans | **1.58** | **11.32** | **2.47** | **11.95** | **111.80** | **1.22** | **3.86** | **20.60** |

Table 3: Hardware utilization on FPGA.

| | $(N,L,d)$ | HD encoder | | | HD transformer | | | HD classifier | | | Total | | |
|---|---|---|---|---|---|---|---|---|---|---|---|---|---|
| | | LUT | FF | BRAM | LUT | FF | BRAM | LUT | FF | BRAM | LUT | FF | BRAM |
| JV | (12,25,128) | 81% | 4% | / | 15% | 12% | 16% | 2% | 1% | 7% | 98% | 18% | 23% |
| HB | (61,405,16) | 54% | 5% | / | 34% | 25% | 55% | 0% | 0% | 1% | 88% | 30% | 56% |
| SAD | (13,93,100) | 59% | 6% | / | 38% | 32% | 38% | 2% | 1% | 6% | 99% | 39% | 44% |
| FD | (144,2,10) | 79% | 4% | / | 4% | 3% | 8% | 0% | 0% | 1% | 83% | 7% | 9% |
| PSF | (963,144,1) | 57% | 5% | / | 7% | 4% | 20% | 0% | 0% | 2% | 65% | 9% | 21% |
| RS | (6,30,200) | 63% | 4% | / | 24% | 20% | 25% | 1% | 1% | 6% | 88% | 25% | 30% |
| Ep | (3,207,80) | 13% | 6% | / | 71% | 57% | 85% | 1% | 0% | 3% | 84% | 64% | 88% |

**Datasets:** JV = JapaneseVowels, HB = Heartbeat, SAD = SpokenArabicDigits, FD = FaceDetection, PSF = PEMS-SF, RS = RacketSports, Ep = Epilepsy.

Meanwhile, the comparison with LeHDC further highlights the effectiveness of the proposed HD Transformer, since both methods employ the same HD classifier, but LeHDC only adopts the vanilla HD spatial-temporal encoding as described in Section 3.2.2. This comparison can thus be regarded as a direct ablation of the HD Transformer component. As shown in Table 1, BiHDTrans consistently achieves substantially higher accuracy than LeHDC across all datasets, with an average improvement of 40.77%, which demonstrates the superiority of the HD Transformer in processing MTS information.

## 6.2 RUNTIME LATENCY

We compare the inference latency of BiHDTrans and SOTA binary Transformers on FPGA, with the clock frequency fixed at 100 MHz. BiHDTrans is implemented with the pipelined design described in Section 5, using a single Transformer encoder block and computing attention only for the final token. For a fair comparison, we also exploit the parallelism of binary Transformers—deploying 64 accumulators for floating-point inputs and 284 accumulators for binary inputs to maximally paralyze the FC layers, and attention scores calculated as the pipeline of BiHDTrans. However, the calculation nonlinear softmax activation, which involves extensive floating-point operations, cannot be further paralyzed due to resource constraints. As shown in Table 2, BiHDTrans achieves at least 39.4× lower latency on average.

While binary Transformers benefit from partial parallelism, their FC layers inherently introduce sequential dependencies. Specifically, the computation of each output neuron requires the complete set of input neuron activations from the preceding layer, as every output is connected to all inputs. This dependency prevents a fully pipelined flow, since the next layer cannot commence until the entire previous layer has been computed. In contrast, BiHDTrans leverages the i.i.d. property of hyperdimensional computing: even when the parallelized dimension $d \ll D$, each dimension can be processed independently, enabling a fully pipelined implementation. This architectural difference allows BiHDTrans to sustain a continuous computation flow across layers and thereby achieve substantial acceleration.

## 6.3 HARDWARE UTILIZATION

We break down the utilization of FPGA resources—look-up tables (LUT), flip-flops (FF), and block RAM (BRAM)—for BiHDTrans. As shown in Table 3, the number of dimensions that can be parallelized ($d$) depends on the multivariate feature size ($N$) and the temporal window length ($L$). When $N > L$, the HD encoder consumes most of the computational resources, as it parallelizes

Table 4: Dimensionality tradeoffs of BiHDTrans.

| Dimension | Japanese-Vowels | Heartbeat | Spoken-Arabic-Digits | Face-Detection | PEMS-SF | Racket-Sports | Epilepsy | Average |
|---|---|---|---|---|---|---|---|---|
| 10,000 | **97.30** | **77.56** | **96.41** | **61.18** | **87.28** | **83.82** | **86.96** | **84.36** |
| 8,100 | 96.76 | 77.07 | 96.32 | 61.04 | 82.66 | 82.24 | 86.23 | 83.19 |
| 6,400 | 97.03 | 76.59 | 96.27 | 59.99 | 82.66 | 73.03 | 84.78 | 81.48 |
| 4,900 | 96.49 | 75.12 | 95.09 | 58.88 | 80.92 | 71.71 | 82.61 | 80.12 |
| 3,600 | 96.76 | 74.15 | 94.72 | 58.51 | 79.19 | 70.39 | 81.88 | 79.37 |
| 2,500 | 95.14 | 74.15 | 93.36 | 57.24 | 77.46 | 61.84 | 80.43 | 77.09 |
| 1,600 | 91.35 | 72.22 | 90.04 | 55.25 | 72.83 | 60.53 | 75.36 | 73.94 |

Table 5: Model size (unit: kB).

| | Japanese-Vowels | Heartbeat | Spoken-Arabic-Digits | Face-Detection | PEMS-SF | Racket-Sports | Epilepsy | Average |
|---|---|---|---|---|---|---|---|---|
| BiBERT | 16.78 | 17.46 | 16.82 | 18.78 | 31.97 | 16.61 | 16.56 | 19.28 |
| BiT | 16.79 | 17.46 | 16.82 | 18.79 | 31.98 | 16.62 | 16.59 | 19.29 |
| BiViT | 16.78 | 17.46 | 16.82 | 18.78 | 31.97 | 16.61 | 16.56 | 19.28 |
| BiHDTrans # | 16.25 | 7.50 | 17.50 | 7.50 | 13.75 | 10.00 | 10.00 | 11.79 |
| BiHDTrans $ | **5.85** | **2.70** | **6.30** | **2.70** | **4.95** | **3.60** | **3.60** | **4.24** |

#: $D = 10000$. \$: $D = 3600$.

over N dimensions. On the other hand, when $N < L$, the HD Transformer utilizes the majority of resources, exploiting parallelism over the temporal window length $L$.

## 6.4 EXPLORING DIMENSIONALITY TRADEOFFS

We further investigate the impact of hyperspace dimensionality on BiHDTrans performance. As shown in Table 4, with a tolerance of 2.5% accuracy loss, the model maintains acceptable performance until the dimensionality is reduced from 8,100 to 6,400. When the dimension decreases from 10,000 to 8,100, BiHDTrans exhibits only a minor accuracy drop of 1.17%, while achieving notable efficiency gains: model size is reduced by 19% and inference latency on FPGA decreases by 17.2%. However, when the dimension is further reduced below 8,100, accuracy degradation exceeds the 2.5% threshold, indicating a clear tradeoff between dimensionality, efficiency, and predictive performance.

Furthermore, BiHDTrans can still outperforms SOTA binary Transformers by 1-2% in accuracy event when dimension drops to 3,600. As shown Table 5, BiHDTrans at this dimensionality achieves an average model size 4.4× smaller than that of SOTA binary Transformers. Compared to the 10,000-dimensional BiHDTrans, the model size is reduced by 64% and the inference latency on FPGA decreases by 49.8%, demonstrating that BiHDTrans can realize a more radical efficiency while remain competitive performance under considerable dimensionality reduction.

## 6.5 EVALUATION ON THE BINARIZED ATTENTION SCORE

We conducted an evaluation of the binarized attention scores on the SpokenArabicDigits dataset. All binarized attention scores are recorded and the average sparsity over all steps within each epoch is computed during the training and validation processes. As illustrated in Figure 3a, the sparsity of the binarized attention remains stable around 0.5 for both training and validation sets. This demonstrates that the binary attention mechanism does not lead to overly sparse or overly dense attention, and thus does not make Transformer training brittle or unstable.

Furthermore, from the perspective of information theory, these observations align with the conclusions of Qin et al. (2022), showing that binarizing attention scores using a Boolean function effectively minimizes information loss. Corresponds to the stable sparsity around 0.5, The maximum output entropy (MOE) of the binary source can be nearly maximized, as shown in Figure 3b, indicating that the binary attention preserves maximal information from the full-precision attention scores.

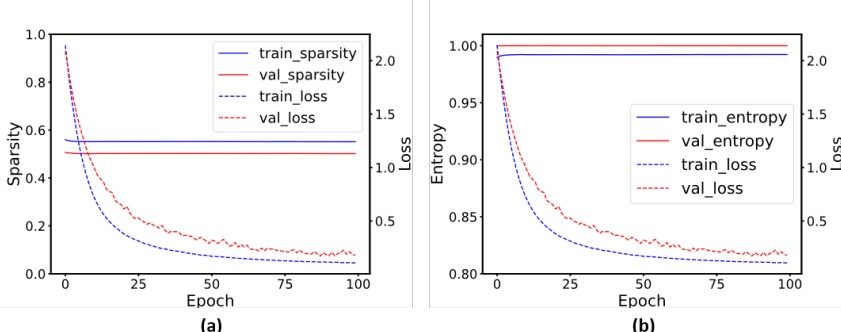

Figure 3: Binarized attention scores of the SpokenArabicDigits dataset: (a) Sparsity. (b) Maximum output entropy.

## 7    CONCLUSION

We present BiHDTrans, a novel neurosymbolic framework that integrates HD computing with Transformers for efficient MTS classification. BiHDTrans achieves superior accuracy and significantly lower latency compared to SOTA HD computing and binary Transformer models. Our work provides both theoretical analysis and empirical results demonstrating the potential of combining the efficiency of HD computing with the powerful modeling capabilities of Transformers, paving the way for accurate and low-latency MTS classification in resource-constrained IoT environments. BiHDTrans can be applied a wide domain where sequential data must be processed under strict efficiency constraints, such as real-time physiological monitoring in wearable healthcare, predictive maintenance in industrial automation, or adaptive control in robotics. Future work could explore scaling BiHDTrans-like algorithm-hardware co-design to efficient large language model inference on FPGA or heterogeneous hardware platforms (Chen et al., 2024).

## 8    ETHICS STATEMENT

In the development and evaluation of BiHDTrans, we have adhered to the ICLR Code of Ethics. This paper does not involve human subjects or sensitive data that require institutional review board (IRB) approval. The datasets used are publicly available and sourced from the latest archive, ensuring compliance with data release practices. We have also taken steps to mitigate potential biases in our models through rigorous validation and testing across diverse datasets. Our research does not aim to generate insights that could be misused or cause harm. We have disclosed all potential conflicts of interest and sponsorships transparently. We affirm our commitment to privacy, security, and legal compliance in all aspects of our research.

## 9    REPRODUCIBILITY STATEMENT

To ensure the reproducibility of our work, we have taken several steps. The detailed methodology, including the architecture of BiHDTrans, training procedures, and experimental setups, is described in the main paper and the appendices. We provide the sources and links of the datasets in the references and source code. Additionally, we have included proofs for our theoretical claims in the appendix to support the validity of our results. To facilitate the reproduction of our experiments, we submit an anonymous link to the source code. We believe these resources will enable other researchers to replicate our findings and build upon our work.

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

# APPENDICES FOR BiHDTRANS

## A   HYPERPARAMETERS OF BiHDTRANS

Table 6: Hyperparameters of BiHDTrans.

| Dataset | HD-dim | Quant lvl. | $D_h$ | Optimizer | LR | WD | Dropout | Batch | Epoch |
|---|---|---|---|---|---|---|---|---|---|
| JapaneseVowels | | | | | 1e-4 | 5e-2 | 0.2 | 4 | 50 |
| Heartbeat | | | | | 1e-5 | 5e-2 | 0.2 | 8 | 100 |
| SpokenArabicDigits | | | | | 1e-6 | 5e-2 | 0.1 | 1 | 100 |
| FaceDetection | 10000 | 256 | 10 | Adam | 1e-6 | 5e-2 | 0.2 | 2 | 100 |
| PEMS-SF | | | | | 1e-5 | 5e-2 | 0 | 1 | 200 |
| RacketSports | | | | | 1e-5 | 5e-2 | 0.1 | 1 | 200 |
| Epilepsy | | | | | 1e-5 | 5e-2 | 0 | 1 | 100 |

The hyperparameters for BiHDTrans are tuned on the validation set and summarized in Table 6. For all datasets, the HD dimension (HD-dim) is set to 10,000 and the number of attention heads ($H_h$) is set to 10, providing a sufficiently high-dimensional space for robust binary hypervector representations. Before HD encoding, all input samples are first normalized to the range [0, 1] using min–max normalization, followed by uniform quantization into 256 discrete levels; the same preprocessing procedure is applied consistently across all datasets. The Adam optimizer is used across all datasets, with batch size from 1 to 8 and learning rates (LR) ranging from 1e-6 to 1e-4 depending on dataset size and complexity, and a consistent weight decay (WD) of 5e-2 to regularize the training. Dropout rates are applied selectively between 0 and 0.2 to prevent overfitting, while the number of training epochs ranges from 50 to 200 to ensure convergence. No temperature parameters or annealing schedules are included for binarization, as training is performed via STE-based method.

## B   PROOFS

In the following proofs, we rely on standard results from high-dimensional probability. For details on all the convergence theorems, concentration inequalities, and other probabilistic results used here, see Vershynin (2018).

### B.1   QUANTIZATION DISTORTION OF BINARIZATION WITH SCALING FACTOR

Assume a random variable $X \sim \mathcal{N}(0, \sigma)$, which is binarized into $B$ with a scaling factor $\varepsilon$:

$$\varepsilon = \mathbb{E}\left[|X|\right] = \int_{-\infty}^{+\infty} |x| \cdot p(x) \, dx = \sigma \sqrt{\frac{2}{\pi}}. \tag{8}$$

The binarization process and corresponding quantization error $e$ are defined as:

$$B = \begin{cases} -\varepsilon, & \text{if } X < 0, \\ \varepsilon, & \text{if } X \geq 0. \end{cases} \tag{9}$$

$$e = X - B. \tag{10}$$

The quantization distortion $D_B$ is given by the expected mean-squared error (MSE) between the original variable and its binarized form:

$$\begin{aligned} D_B = E[e^2] &= E[(X - B)^2] \\ &= \int_0^\infty (x - \varepsilon)^2 \cdot p(x) \, dx + \int_{-\infty}^0 (x + \varepsilon)^2 \cdot p(x) \, dx \\ &= \sigma^2 + \varepsilon^2 - 2\varepsilon \cdot E[|x|] \\ &= \sigma^2 \left(1 - \frac{2}{\pi}\right). \end{aligned} \tag{11}$$

B.2 QUANTIZATION DISTORTION OF BINARIZATION IN HYPERSPACE

Before HD mapping, the variable X is first quantized into q discrete levels. Let the quantized variable be denoted as $X_q$. Assume $X \sim \mathcal{N}(0, \sigma)$, and the quantizer has output range $[-L, L]$. Setting $L = 3\sigma$ covers approximately 99.7% of the data. The quantization step size is then:

$$d = \frac{2L}{q} = \frac{6\sigma}{q}. \tag{12}$$

For a uniform quantizer, the quantization error within each interval can be approximated as uniformly distributed in $[-\frac{d}{2}, \frac{d}{2}]$. Thus, the quantization distortion $D_Q$ is:

$$D_Q = \frac{d^2}{12} = \frac{3\sigma^2}{q^2}. \tag{13}$$

For each quantized value $x_i \in X_q$, it is treated as belonging to a distinct position domain $p_i$. During HD mapping, $x_i$ and $p_i$ are independently mapped into two sets of binarized seed hypervectors, namely the value hypervector and the position hypervector (as described in Section 3.2.1):

$$\begin{cases} \text{value: } x_i \to X_i, & i = 1, 2, \ldots, k \\ \text{position: } p_i \to P_i, & i = 1, 2, \ldots, k \end{cases}. \tag{14}$$

After hash-table encoding (Heddes et al., 2023) and binarization, the combined hypervector is:

$$\begin{aligned} V &= X_1 \odot P_1 + X_2 \odot P_2 + \cdots + X_k \odot P_k, \\ V_b &= [V] = [X_1 \odot P_1 + X_2 \odot P_2 + \cdots + X_k \odot P_k], \end{aligned} \tag{15}$$

where $[\cdot]$ denotes the binarization operation. $V_b$ is a holistic holographic representation of all information. Decoding a particular component can be performed as:

$$\begin{aligned} X_i' &= V_b \odot P_i = [X_1 \odot P_1 + X_2 \odot P_2 + \cdots + X_k \odot P_k] \odot P_i \\ &= [X_1 \odot P_1 \odot P_i + X_2 \odot P_2 \odot P_i + \cdots + X_k \odot P_k \odot P_i] \\ &= [X_i + \text{noise}]. \end{aligned} \tag{16}$$

By computing $\cos(X_i', X_i)$, the most similar value hypervector can be retrieved, yielding the decoded quantized value $x_i'$. Due to the sparsity and robustness of high-dimensional spaces, the noise can be effectively neglected, giving:

$$X_i' = [X_i + \text{noise}] \approx [X_i] = X_i. \tag{17}$$

Thus, in this case, the distortion introduced by binarization after HD hash-table encoding is negligible.

Generalizing this principle, the quantization distortion of binarizing any full-precision hypervector $S$ can be defined as the difference in similarity with respect to a particular decoding hypervector $X_d$, that is:

$$D_H := |\cos(S, X_d) - \cos(S_{bin}, X_d)|, \tag{18}$$

where $S_{bin} = [S]$. This definition is more principled than simply using $D_H := \cos(S, S_{bin})$, since the direct distance between $S$ and $S_{bin}$ may not correspond to an actual distortion in the real-valued domain, thanks to the sparsity and robustness of hyperspace (Thomas et al., 2022).

B.3 PROOF OF THEOREM 1

**Theorem 1.** *Let $X \sim \mathcal{N}(0, \sigma)$, and the dimension of the hyperspace $D \to \infty$. Given that the quantization level $q \geq 3$ for the real-valued feature, the information distortion from binarizing $X$ in hyperspace is lower than that incurred by directly binarizing the real-valued data.*

*Proof.* HD computing begins with the random generation of seed hypervectors (e.g., position and value hypervectors) whose elements are i.i.d. binary variables taking values -1 or +1 with equal probability. Since HD operations preserve this distributional property, any non-binary hypervector that arises during HD computation can be regarded as the aggregation of such i.i.d. binary hypervectors.

Let $H_1, H_2, \ldots, H_N \in \{-1, +1\}^D$ denotes $N$ i.i.d. binary hypervectors, where each component $h_{ji}$ satisfies $\mathbb{P}(h_{ji} = +1) = \mathbb{P}(h_{ji} = -1) = 0.5$. Define the aggregated hypervector

$$S = \sum_{j=1}^{N} H_j = (S_1, S_2, \ldots, S_D), \tag{19}$$

where each component is given by

$$S_i = \sum_{j=1}^{N} h_{ji}. \tag{20}$$

We have the linearity of expectation and variance:

$$\mathbb{E}[S_i] = \mathbb{E}\left[\sum_{j=1}^{N} h_{ji}\right] = \sum_{j=1}^{N} \mathbb{E}[h_{ji}] = 0, \tag{21}$$

$$\mathrm{Var}(S_i) = \mathrm{Var}\left[\sum_{j=1}^{N} h_{ji}\right] = \sum_{j=1}^{N} \mathrm{Var}[h_{ji}] = N. \tag{22}$$

Thus, each $S_i$ is i.i.d. with zero mean (symmetrical) and finite second moment. By the central limit theorem, $S_i \sim \mathcal{N}(0, N)$. Hence, $|S_i|$ follows a folded Gaussian distribution, yielding

$$\mathbb{E}[|S_i|] = \sqrt{\frac{2N}{\pi}}, \tag{23}$$

$$\mathrm{Var}(|S_i|) = N\left(1 - \frac{2}{\pi}\right). \tag{24}$$

By the law of large numbers, the sample mean converges in probability to its expectation:

$$\frac{1}{D} \sum_{i=1}^{D} |S_i| \xrightarrow{P} \mathbb{E}[|S_i|] = \sqrt{\frac{2N}{\pi}}. \tag{25}$$

A sub-Gaussian tail bound further gives the deviation probability

$$\mathbb{P}\left(||S_i| - \mathbb{E}[|S_i|]| > \epsilon\right) \le 2\exp(-\frac{\epsilon}{2\mathrm{Var}(|S_i|)}) = 2\exp(-\frac{\epsilon}{2N\left(1 - \frac{2}{\pi}\right)}), \tag{26}$$

and a Bernstein–Chernoff bound shows that the fraction $\delta$ of components violating this deviation decays exponentially:

$$\mathbb{P}\left(\frac{1}{D} \sum_{i=1}^{D} \math1\{||S_i| - \mathbb{E}[|S_i|]| > \epsilon\} > \delta\right) \le 2e^{-c'D}, \quad \text{where } c' = \Theta\left(\frac{\epsilon^2}{N}\right). \tag{27}$$

Therefore, as $D \to \infty$, with high probability most components $|S_i|$ lie in the interval $\left[\sqrt{\frac{2N}{\pi}} - \epsilon, \sqrt{\frac{2N}{\pi}} + \epsilon\right]$, and the concentration strengthens exponentially, which indicates the amplitude concentration of $S$

$$|S_i| \xrightarrow{D \to \infty} \sqrt{\frac{2N}{\pi}} =: c. \tag{28}$$

We now compute $D_H$ according to (18):

$$\begin{aligned} D_H &= |\cos(S, X_d) - \cos(S_{\text{bin}}, X_d)| \\ &= \left|\frac{S \cdot X_d}{\|S\| \times \sqrt{D}} - \frac{S_{bin} \cdot X_d}{\sqrt{D} \times \sqrt{D}}\right| \\ &= \frac{X_d}{\sqrt{D}} \left|\frac{S}{\|S\|} - \frac{S_{bin}}{\sqrt{D}}\right|. \end{aligned} \tag{29}$$

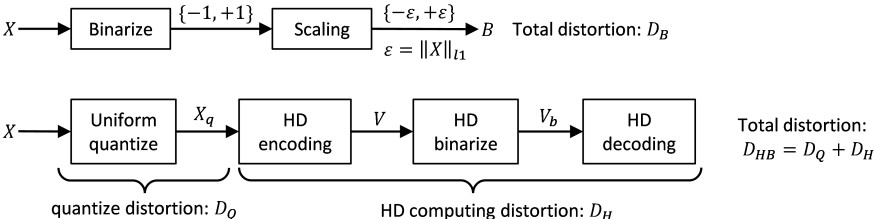

Figure 4: Process of direct binarization (upper) and binarization in hyperspace (lower).

Let $\Delta = \left| \frac{S}{\|S\|} - \frac{S_{\text{bin}}}{\sqrt{D}} \right|$ and square expansion yields

$$\Delta^2 = \sum_{i=1}^{D} \left( \frac{S_i}{\|S\|} - \frac{\text{sign}(S_i)}{\sqrt{D}} \right)^2 = 2 \left( 1 - \frac{1}{\|S\|\sqrt{D}} \sum_{i=1}^{D} |S_i| \right). \tag{30}$$

By Slutsky's theorem, convergence in probability is preserved under continuous mappings. Hence, as $D \to \infty$,

$$\sum_{i=1}^{D} |S_i| \xrightarrow{D \to \infty} D \cdot c, \tag{31}$$

$$\|S\|^2 = \sum_{i=1}^{D} S_i^2 = \sum_{i=1}^{D} |S_i|^2 \xrightarrow{D \to \infty} D \cdot c^2, \tag{32}$$

$$\|S\| \xrightarrow{D \to \infty} \sqrt{D} \cdot c, \tag{33}$$

and thus

$$\frac{1}{\|S\|\sqrt{D}} \sum_{i=1}^{D} |S_i| \xrightarrow{D \to \infty} \frac{D \cdot c}{\sqrt{D} \cdot c \cdot \sqrt{D}} = 1, \tag{34}$$

$$\Delta^2 \xrightarrow{D \to \infty} 0 \implies D_H \xrightarrow{D \to \infty} 0. \tag{35}$$

That is, the hyperspace binarization distortion vanishes exponentially fast as dimension grows.

As shown in Figure 4, the total quantization distortion of HD encoding followed by hyperspace binarization is

$$D_{HB} = D_Q + D_H. \tag{36}$$

Since $D_H \approx 0$ when $D$ is large, we have $D_{HB} \approx D_Q$. To ensure $D_{HB} < D_B$, it suffices that $D_Q < D_B$. From (11) and (13), this condition reduces to

$$\frac{3\sigma^2}{q^2} < \sigma^2 \left( 1 - \frac{2}{\pi} \right) \implies q > \sqrt{\frac{3}{1 - \frac{2}{\pi}}} \approx 2.87. \tag{37}$$

Hence, when the quantization level satisfies $q \geq 3$, the theorem holds.

### B.4 PROOF OF THEOREM 2

**Theorem 2.** *Let $\{V_i\}_{i=1}^{N} \in \{-1, +1\}^D$ be a set of binary hypervectors, and let $w = (w_1, w_2, \ldots, w_N) \in \mathbb{R}^N$ be a set of real-valued weights. The weighted sum with real-valued and binarized weights are given by: $Y = \text{sign}\left( \sum_{i=1}^{N} w_i \cdot V_i \right)$ and $Y' = \text{sign}\left( \sum_{i=1}^{N} w_{qi} \cdot V_i \right)$, where $w_{qi} \in 0, 1$ is the binarized weight (mask). For any fixed $\alpha \in (0, \frac{1}{2})$, there exists a constant $C(w, \alpha) > 0$ such that as $D \to \infty$, the information distortion between $Y$ and $Y'$ is bounded by $C(w, \alpha)$ and converges to zero with high probability.*

*Proof.* Since hypervectors are i.i.d. across dimensions, consider the *j*-th component of the weighted sum:

$$S_j = \sum_{i=1}^{N} w_i \cdot V_{ij}, \quad S_j' = \sum_{i=1}^{N} w_{qi} \cdot V_{ij} \tag{38}$$

Here, $V_{ij} \in \{-1, +1\}$ are i.i.d., with $\mathbb{E}[V_{ij}] = 0$, $\mathrm{Var}(V_{ij}) = 1$. Hence,

$$\mathbb{E}[S_j] = \sum_{i=1}^{N} w_i \mathbb{E}[V_{ij}] = 0, \quad \mathrm{Var}(S_j) = \sum_{i=1}^{N} w_i^2 \mathrm{Var}(V_{ij}) = \sum_{i=1}^{N} w_i^2 =: \sigma_S^2, \tag{39}$$

$$\mathbb{E}[S_j'] = \sum_{i=1}^{N} w_{qi} \mathbb{E}[V_{ij}] = 0, \quad \mathrm{Var}(S_j') = \sum_{i=1}^{N} w_{qi}^2 = \sum_{i:w_i>0} 1 = N_+. \tag{40}$$

By the central limit theorem, we approximate $S_j \sim \mathcal{N}(0, \sigma_S^2)$, $S_j' \sim \mathcal{N}(0, N_+)$.

Define $\Delta = Y - Y' \in \{-2, 0, 2\}^D$. For the $j$-th dimension, $\Delta_j \neq 0$ if and only if $Y_j \neq Y_j'$. Let $r := \frac{1}{D} \sum_{j=1}^{D} \mathbb{K}[Y_j \neq Y_j']$ denotes the disagreement rate between $Y$ and $Y'$. According to (18), the information distortion between $Y$ and $Y'$ is $D_W := |\cos(Y, X_d) - \cos(Y', X_d)|$. Hence,

$$\|Y - Y'\|^2 = \sum_{j=1}^{D} \Delta_j^2 = 4Dr \implies \|Y - Y'\| = 2r\sqrt{D}. \tag{41}$$

$$D_w = \frac{1}{D}|(Y - Y') \cdot X_d| \leq \frac{\|\Delta\|\|X_d\|}{D} = 2\sqrt{r}. \tag{42}$$

Let $Z_j := \mathbb{K}[Y_j \neg Y_j'] \in \{0, 1\}$, which are i.i.d. Bernoulli with $\mathbb{E}[Z_j] = \mathbb{P}(Y_j \neq Y_j') =: p$. Then $r = \frac{1}{D} \sum_{j=1}^{D} Z_j$ is their empirical mean by the law of large numbers such that $r \xrightarrow{a.s.} p$. With deviation probability $\epsilon = D^{-\alpha}$ and $\alpha \in (0, \frac{1}{2})$, Hoeffding's inequality gives that when $D \to \infty$, with high probability $r$ concentrates around $p$ such that

$$\mathbb{P}\left(|r - p| \geq D^{-\alpha}\right) \leq 2\exp(-2D^{1-2\alpha}) \to 0. \tag{43}$$

Now define $\delta_j := S_j - S_j' = \sum_{i=1}^{N}(w_i - w_{qi}) \cdot V_{ij}$, A sufficient condition for sign flip $Y_j \neq Y_j'$ is

$$Y_j \neq Y_j' \implies |\delta_j| \geq |S_j'|. \tag{44}$$

Thus,

$$p = \mathbb{P}(Y_j \neq Y_j') \leq \mathbb{P}(|\delta_j| \geq |S_j'|) \leq \mathbb{P}(|\delta_j| \geq \varepsilon) + \mathbb{P}(|S_j'| \leq \varepsilon), \tag{45}$$

for any $\varepsilon > 0$. For $|\delta_j|$, Hoeffding's inequality controls the upper bound of $\mathbb{P}(|\delta_j| \geq \varepsilon)$, which yields

$$\mathbb{P}(|\delta_j| \geq \varepsilon) \leq 2\exp\left(-\frac{\varepsilon^2}{2\sum_{i=1}^{N}(w_i - w_{qi})^2}\right) \leq 2\exp\left(-\frac{\varepsilon^2}{2\sum_{i=1}^{N} w_i^2}\right). \tag{46}$$

For $|S_j'|$, $\mathbb{P}(|S_j'| \leq \varepsilon)$ is controlled by Gaussian tail bounds:

$$\mathbb{P}(|S_j'| \leq \varepsilon) \leq \int_{-\varepsilon}^{\varepsilon} \frac{1}{\sqrt{2\pi N_+}} \exp(-\frac{x^2}{2N_+})\, dx \leq \frac{2\varepsilon}{\sqrt{2\pi N_+}}. \tag{47}$$

Taking $\varepsilon = D^{-\alpha}$ with $\alpha \in (0, \frac{1}{2})$, as $D \to \infty$, according to (45), (46), and (47), we have

$$p \leq 2\exp\left(-\frac{D^{-2\alpha}}{2\sum_{i=1}^{N} w_i^2}\right) + \frac{2D^{-\alpha}}{\sqrt{2\pi N_+}} \to 0. \tag{48}$$

Combining (42), (43), and (48), we concluded that $D_w \to 0$ with high probability as $D \to \infty$. If define,

$$C(w, \alpha) := 2\sqrt{2\exp\left(-\frac{\varepsilon^2}{2\sum_{i=1}^{N} w_i^2}\right) + \frac{2\varepsilon}{\sqrt{2\pi N_+}} + \epsilon}, \tag{49}$$

we have

$$\mathbb{P}(D_w > C(w, \alpha)) \leq 2\exp(-2D^{1-2\alpha}) \to 0, \quad \text{as } D \to \infty. \tag{50}$$

Therefore, the information distortion is bounded by $C(w, \alpha)$ and vanishes with high probability in the high-dimensional limit.

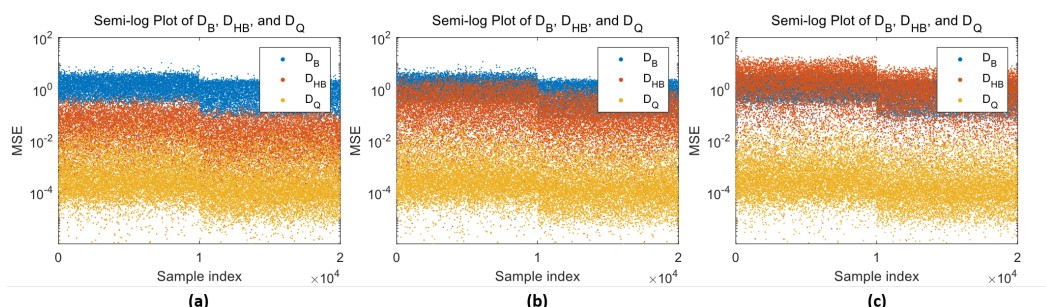

Figure 5: Empirical evaluation of Theorem 1. Dimensionality: **(a)** 10,000. **(b)** 3,600. **(c)** 900.

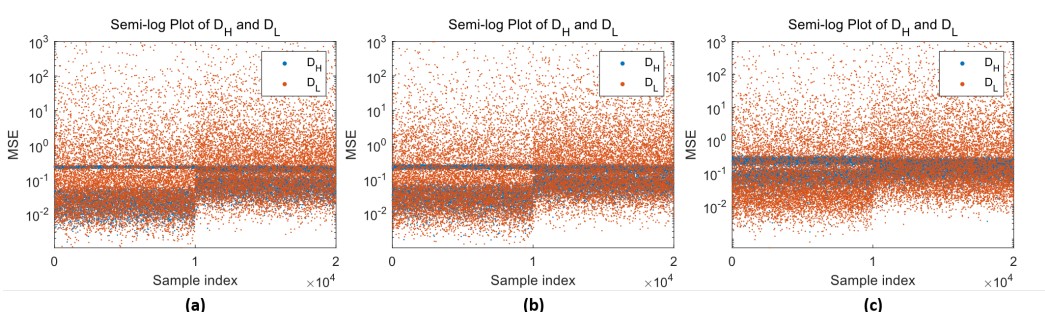

Figure 6: Empirical evaluation of Theorem 2. Dimensionality: **(a)** 10,000. **(b)** 3,600. **(c)** 900.

## C  ADDITIONAL RESULTS

### C.1  EMPIRICAL VALIDATION OF THEOREM 1

We empirically validate Theorem 1 using randomly generated multivariate variable $X$. As shown in Figure 4, we consider two cases: (i) directly binarizing $X$ and computing $D_B$ as described in Appendix B.1, and (ii) mapping, encoding, and binarizing $X$ in hyperspace, followed by decoding back to real-valued $X'$ using the simple hash-table encoding in Appendix B.2. Since $X'$ can be explicitly reconstructed in this setting, $D_{HB}$ is calculated as the MSE between $X$ and $X'$. We also compute $D_Q$ as described in Appendix B.2, serving as an intermediate measure.

To ensure robustness, $X$ is generated 20,000 times: the first 10,000 samples from a Gaussian distribution and the remaining 10,000 from a uniform distribution. The detailed experimental setups are summarized in Table 7. The results, presented in Figure 5, show that $D_{HB}$ is on average  31× smaller than $D_B$, with  14× less standard deviation when dimensionality $D = 10,000$ (Figure 5a). However, when dimensionality reduces to 3,600, although $D_{HB}$ is still smaller than $D_B$ on average, the margin has become much less significant, as shown in Figure 5b. When dimensionality subsequently reduces to 900, the high-dimensional limit can no longer be tenable. As shown in Figure 5c, the average value of $D_{HB}$ has become larger than that of $D_B$. These results provide strong empirical support for Theorem 1.

### C.2  EMPIRICAL VALIDATION OF THEOREM 2

We extend the validation to Theorem 2 by generating sets of multivariate variables $\boldsymbol{X} = [X_1, X_1, \ldots, X_L]$ with associated weights $\boldsymbol{w} = [w_1, w_1, \ldots, w_L]$ for weighted additions. The real-valued weights are binarized into 0/1 masks $\boldsymbol{w_q}$, where position values are set to 1 and negative values to 0. Accordingly, the weighted mean is defined as $W = \boldsymbol{X^T w}/\sum_i w_i$ and the masked mean as $W' = \boldsymbol{X^T w_q}/\sum_i w_{qi}$. The information distortion of binarizing weights in the real domain is measured by the MSE between $W$ and $W'$, denoted as $D_L$.

Table 7: Experimental setups for empirical evaluation of Theorem 1.

| Distribution | $\sigma$ | $a$ | Quant lvl. | Channel No. | $D$ |
|---|---|---|---|---|---|
| $X \sim \mathcal{N}(0, \sigma)$ | Uniform$(1, 3)$ | / | Uniform$(16, 256)$ | Uniform$(2, 100)$ | 10000 |
| $X \sim \mathcal{U}(-a, +a)$ | / | Uniform$(1, 5)$ | | | |

Table 8: Experimental setups for empirical evaluation of Theorem 2.

| Distribution | Quant lvl. | Channel No. | $L$ | $D$ |
|---|---|---|---|---|
| $X \sim \mathcal{N}(0, 1)$ | 256 | Uniform$(10, 100)$ | 100 | 10000 |
| $X \sim \mathcal{U}(-3, +3)$ | | | | |

In parallel, $\boldsymbol{X}$ is also mapped to hyperspace following Appendix B.2. Weighted and masked means of the hypervectors, denoted as $Y$ and $Y'$, are computed according to Theorem 2 and then decoded back to real values as described in Appendix B.2. The distortion of binarizing weights in hyperspace, denoted as $D_H$, is obtained as the MSE between the decoded $Y$ and $Y'$.

As in Appendix C.1, we generate 20,000 random samples, with the first 10,000 from a Gaussian distribution and the remaining 10,000 from a uniform distribution. Detailed experimental setups are given in Table 8. Results in Figure 6 show that although $D_H$ is not always smaller than $D_L$, it is substantially more stable and consistently bounded, avoiding catastrophic distortions. Meanwhile, the overall trend in the distance statistics remains essentially unchanged across 10,000, 3,600, and 900 dimensions. This behavior can be attributed to the fact that the hypervectors remain i.i.d. regardless of their dimensionality, and this i.i.d. property is precisely the key assumption under which Theorem 2 holds. In other words, although reducing the dimensionality inevitably increases the information distortion of an individual hypervector under binarization (as reflected in Figure 5), it does not necessarily cause a significant change in the information distortion introduced when binarizing the weights during weighted summation, which is what Theorem 2 characterizes.

In Theorem 2, according to asymptotic limit argument, the theoretical upper bound decreases as dimensionality increases. When the dimensionality is reduced, the upper bound increases to some extent, and the system's error tolerance correspondingly decreases. Empirically, this manifests as a higher proportion of $D_H$ values approaching the theoretical upper bound at lower dimensions. The empirical results presented in Figure 6 align with this analysis. These observations provides a solid verification of Theorem 2.

### C.3 EVALUATION ON THE SENSITIVITY OF QUANTIZATION LEVEL

To further investigate the robustness of our value encoding scheme, we evaluate BiHDTrans under different quantization levels for the value hypervectors, as summarized in Table 9. The results indicate that BiHDTrans exhibits robustness with respect to the quantization level used in the HD value encoding. This behavior is theoretically well-grounded from various perspectives.

First, within HD computing paradigm, information is distributed across a large number of independent dimensions. Such distributed representations inherently dilute quantization noise, allowing coarse quantization to retain the essential statistical relationships among hypervectors (Thomas et al., 2022). This aligns with prior HD computing studies, where the quantization level has been shown to exert limited influence on model's performance and typically not treated as a critical hyperparameter, provided it lies within a reasonable range determined by dataset characteristics (Rahimi et al., 2019).

Additionally, from the neural network perspective, many prior studies have demonstrated that well-parameterized neural networks can tolerate moderate quantization without jeopardizing performance (Frankle & Carbin, 2019). Since the neuro-symbolic BiHDTrans can be viewed as a special form of neural network (trained as an equivalent BNN), its robustness to quantization is also consistent with these findings.

Furthermore, these results also validate the theoretical analysis: mapping into hyperspace makes subsequent binarization more robust to quantization loss than direct weight binarization and binary activation. Even when the quantization level is reduced to 16, the average accuracy of BiHDTrans still surpasses the SOTA binary Transformer by 4.8%, which is a direct confirmation to Theorem 1.

Table 9: The sensitivity of quantization level of BiHDTrans.

| Quant. level | Japanese-Vowels | Heartbeat | Spoken-Arabic-Digits | Face-Detection | PEMS-SF | Racket-Sports | Epilepsy | Average |
|---|---|---|---|---|---|---|---|---|
| 256 | **97.30** | **77.56** | **96.41** | 61.18 | **87.28** | **83.82** | **86.96** | **84.36** |
| 128 | 96.76 | 75.61 | 96.36 | 60.61 | 86.82 | 83.47 | **86.96** | 84.36 |
| 64 | 97.03 | 75.61 | 96.13 | **61.95** | 84.39 | 82.66 | 86.23 | 83.80 |
| 32 | 95.68 | 75.61 | 96.04 | 60.90 | 84.97 | 80.92 | **86.96** | 83.43 |
| 16 | 96.49 | 75.12 | 95.59 | 59.76 | 82.66 | 81.58 | 86.23 | 82.49 |

Table 10: Comparison to SOTA binary Transformers for anomaly detection.

| | SMAP | | | MSL | | |
|---|---|---|---|---|---|---|
| | Precision | Recall | $F_{0.5}$ Score | Precision | Recall | $F_{0.5}$ Score |
| Full Precision | *0.592* | *1.000* | *0.645* | *0.813* | *0.929* | *0.833* |
| BiBERT | 0.511 | 0.691 | 0.539 | 0.719 | 0.885 | 0.747 |
| BiT | 0.513 | 0.357 | 0.471 | 0.710 | 0.846 | 0.733 |
| BiViT | 0.513 | 0.357 | 0.471 | 0.735 | **0.962** | 0.772 |
| BiHDTrans | **0.529** | **0.945** | **0.580** | **0.774** | 0.961 | **0.805** |

## C.4 EXTENSION TO ANOMALY DETECTION

To further assess the generality of BiHDTrans, we extend it to anomaly detection task. Since HD computing, and consequently BiHDTrans, is inherently designed for supervised classification, anomaly detection is realized using a one-class classification formulation, which requires only minimal modifications to the model. Specifically, the model is trained exclusively on normal samples. For our implementation, during training, the normal sequential samples are encouraged to cluster a normality center, which is determined as the centroid of all normal samples. At inference time, a test sample is regarded as anomalous if its representation deviates from this centroid by more than a pre-defined threshold. Notably, for purely NN-based full precision model and SOTA binary Transformers, the distance between samples and the centroid is determined by Euclidean distance, and are trained based on MSE loss. In contrast, for BiHDTrans, the distances are measured by cosine similarity and is trained by cosine distance loss, as all samples are in hypervector form.

The experiments are conducted on the NASA anomaly detection datasets SMAP and MSL (Hundman et al., 2018). For these two datasets, precision is a more critical metric than recall rate, as false alarms are particularly costly. Therefore, $F_{0.5}$ score is suggested to be adopted as the primary metrics. As shown in Table 10, BiHDTrans achieves at least 0.041 and 0.058 higher $F_{0.5}$ scores on SMAP and MSL dataset compare to SOTA binary Transformer, respectively. This consistency with the classification results in Table 1 demonstrates that the proposed BiHDTrans generalizes effectively beyond classification and offers a compact, efficient solution for a more general time-series representation.

