# OpenReview forum: "BiHDTrans: binary hyperdimensional transformer for efficient multivariate time series classification"
_ICLR.cc/2026/Conference — Submitted to ICLR 2026_

### Official Review · Reviewer_mTvZ · 2025-10-27

**Soundness:** 2
**Presentation:** 2
**Contribution:** 2
**Rating:** 4
**Confidence:** 3

**Summary:**

The paper introduces BiHDTrans, which is a binary hyper-dimensional transformer for multi-variate time series classification. The paper designs an HD encoder that converts real-valued features into binary HD vectors. The paper then modifies the transformer into a carefully designed binary version. Finally the paper implements the architecture on FPGA and achieves efficient inference.

**Strengths:**

The paper is technically challenging. Both training a binary transformers and deployment on FPGA are non-trivial and have significant contributions. If the entire process is reproducible, it could potentially have big impact on deploying LLM on edge devices.

**Weaknesses:**

* The selected tasks and datasets in the paper are quite simple. From Table 1, the small performance gap between full precision and the proposed method may be caused by the datasets are too simple. For Epilepsy dataset, the proposed method has a significant performance drop compared with full precision, which may hint the proposed method suffers from under performance.
* The proposed method has limited compatibility. It looks very difficult to add any additional transformers features, such as rope or alibi to the proposed  transformer.

**Questions:**

It sounds to me deploying a LLM using the proposed method has much more potential than multi-variate time series classification. What prevents this from happening?

---

> ### Author Response · Authors · 2025-11-22
> **Response to all questions -- Part 1**
>
> Thank you for your comments. Below, we respond to all questions.
>
> -	*It sounds to me deploying a LLM using the proposed method has much more potential than multi-variate time series classification. What prevents this from happening?*
>
> We agree that scaling BiHDTrans to LLMs is a compelling next frontier. From a technical standpoint, extending our HD-Transformer architecture to LLM-scale models is entirely feasible. The reason we first targeted multivariate time-series classification (MTSC) is because we wanted to establish a **minimum viable demonstration** in a problem setting with a more controllable scale. Specifically, MTSC is a representative edge-AI application whose manageable model size allowed us to focus on validating the core value of our neuro-symbolic design, namely, the feasibility of performing self-attention entirely within the binary HDC paradigm. On the hardware side, this also enabled us to implement a fully pipelined design with an entry-level Xilinx Artix-7 FPGA, which verifies the efficiency of the i.i.d. dimension-wise parallelism.
>
> We believe that demonstrating technical feasibility and effective algorithm–hardware co-design on MTSC establishes a critical foundation for scaling BiHDTrans to LLMs. With this foundation, the next-stage work, such as an efficient FPGA-accelerated neuro-symbolic LLM, can focus on the real LLM-specific challenges (without worrying about fundamental issues such as whether BiHDTrans truly reduces information distortion during binarization or whether the design remains hardware-friendly and efficient): how to distill knowledge from pretrained models into a neuro-symbolic architecture, how to allocate compute/memory resources and memory port/bandwidth constraints (Chen et al., 2024).
>
> Therefore, we believe MTSC is not as a limitation, but as a necessary stepping stone that ensures future LLM-scale deployment is both technically grounded and practically achievable.
>
> -	*The proposed method has limited compatibility. It looks very difficult to add any additional transformers features, such as rope or alibi to the proposed transformer.*
>
> We would like to kindly clarify that the proposed BiHDTrans design is not inherently incompatible with positional encoding mechanisms such as RoPE or ALiBi. In fact, if applying this neuro-symbolic Transformer architecture to LLM-style models, a practical approach would be to first apply positional encoding in the real-valued embedding space as usual, and then map the resulting positional-encoded vectors into hypervectors. After this mapping step, all subsequent computations can proceed in the hypervector domain using element-wise operations.
>
> -	*The selected tasks and datasets in the paper are quite simple.*
>
> We acknowledge that some UEA archive datasets could be rather simple. Meanwhile, we would also like to kindly expound that except of the InsectWingbeat dataset, where almost all SOTA models seems to failed to realize validate classification (Eldele et al., 2024), we have utilized 2 of 3 largest datasets in the UEA archive (SpokenArabicDigits, FaceDetection). In addition, we have extended our evaluation to anomaly detection, utilizing the NASA datasets SMAP and MSL, to validate the model’s generality in **Table 10** in **Appendix C.4 Extension to anomaly detection (line 1144)**. The results align with the MTSC results in Table 1, demonstrating that the proposed BiHDTrans generalizes effectively beyond classification and offers a compact, efficient solution for a more general time-series representation.
>
> **Table 10: Comparison to SOTA binary Transformers for anomaly detection.**
> | Model          | SMAP Precision | SMAP Recall | SMAP F0.5 Score | MSL Precision | MSL Recall | MSL F0.5 Score|
> |----------------|----------------|-------------|-----------------|---------------|-----------|----------------|
> | Full Precision | *0.592*        | *1.000*     | *0.645*         | *0.813*       | *0.929*   | *0.833*        |
> | BiBERT         | 0.511          | 0.691       | 0.539           | 0.719         | 0.885     | 0.747          |
> | BiT            | 0.513          | 0.357       | 0.471           | 0.710         | 0.846     | 0.733          |
> | BiViT          | 0.513          | 0.357       | 0.471           | 0.735         | **0.962** | 0.772          |
> | BiHDTrans      | **0.529**      | **0.945**   | **0.580**       | **0.774**     | 0.961     | **0.805**      |

---

> ### Author Response · Authors · 2025-11-23
> **Response to all questions -- Part 2**
>
> -	*For Epilepsy dataset, the proposed method has a significant performance drop compared with full precision, which may hint the proposed method suffers from under performance.*
>
> We would like to kindly clarify that the primary focus of this work is *not* to match the performance of full-precision Transformers with a fully binary model. From a mathematical perspective, achieving near–full-precision accuracy under extreme binarization is fundamentally difficult due to the severe information bottleneck. Instead, the goal of BiHDTrans is to leverage key properties of the high-dimensional (HD) binary space to design a Transformer module that is simultaneously hardware-friendly and more effective than existing binary alternatives. Specifically: (1) compared with SOTA binary Transformers, BiHDTrans consistently delivers higher accuracy, smaller model size, and lower computational cost across the evaluated datasets; (2) compared with inherently efficient HD computing models, BiHDTrans exhibits substantially stronger sequential modeling capability.
>
> In summary, while a performance gap to full precision is expected, the results confirm that BiHDTrans achieves the goals of this work: providing a compact, hardware-friendly, and significantly more effective binary Transformer module than existing HD or binary Transformer baselines.

---

### Official Review · Reviewer_7whK · 2025-10-28

**Soundness:** 3
**Presentation:** 3
**Contribution:** 3
**Rating:** 6
**Confidence:** 4

**Summary:**

This paper proposes a binary hyperdimensional transformer for multivariate time series (MTS) classification with FPGA acceleration. The model changes the conventional transformer computation (primarily the attention mechanisms) to HDC operations. Theoretical results show that binarizing the vector after projecting to high-dimensional holographic space loses less information than binarizing neural networks directly, and binarizing attention scores introduces information distortion that diminishes at higher vector dimensions. The algorithm, including an FPGA implementation of the pipeline, is evaluated against binary neural network models and other HDC models, showing competitive or improved accuracy and significantly better inference latency.

**Strengths:**

1. To the best of the reviewer's knowledge, this is the first work that fully integrates an attention mechanism within the HDC paradigm, improving the paper's originality.
2. The paper provides strong theoretical support for the validity of binarization in BiHDTrans, which is further validated by empirical results.
3. The paper demonstrates a practical path to fully binary, low-latency MTS classification with measured FPGA speedups (40×) while retaining accuracy competitive with full-precision one-block Transformers on these tasks. This directly targets IoT/edge deployment constraints.

**Weaknesses:**

1. Performance evaluation: While the experiment is run on several datasets, most of them are small. Additional datasets of a larger scale are recommended to stress-test the model, such as PTB-XL and Sleep-EDF.
2. Ablation studies and sensitivity analysis: there is only one sensitivity analysis on the vector dimension. see questions for details.

**Questions:**

1. Replacing softmax with a hard binary mask removes temperature/normalization and can typically make transformer training brittle or overly sparse/dense depending on scale. What's the effective sparsity of $Ba$ during training and inference?
2. What is the quantization level used for each dataset, and what quantization method is used? How sensitive is the performance with respect to quantization parameters?
3. Using a different font for (or italicizing) the accuracies of full-precision model in Table $1$ is recommended, because the bolded results seem to be the best model performance excluding those of the full-precision model.

---

> ### Author Response · Authors · 2025-11-22
> **Response to all questions -- Part 1**
>
> Thank you for your comments. Below, we respond to all questions.
>
> -	*Replacing softmax with a hard binary mask removes temperature/normalization and can typically make transformer training brittle or overly sparse/dense depending on scale. What's the effective sparsity of $B_a$ during training and inference?*
>
> We have reported and analyzed the sparsity of $B_a$ during training and inference, as shown in **Figure 3** in **Section 6.5 Evaluation on the Binarized Attention Score (line 475)**. The sparsity of $B_a$ remains stable around 0.5 for both training and validation sets, thus does not lead to overly sparse or overly dense attention and make Transformer training brittle or unstable. Additionally, we would like to highlight that the feasibility of replacing softmax with a binary attention mask was first investigated in BiBERT (Qin et al., 2022) from the perspective of information theory. Therefore, we also include the maximum output entropy (MOE) of $B_a$ in **Figure 3** to further support the stability and effectiveness of our design.
>
> -	*What is the quantization level used for each dataset, and what quantization method is used? How sensitive is the performance with respect to quantization parameters?*
>
> The quantization level for all datasets are fixed to 256, with uniform quantization after normalizing to [0,1]. Additionally, we analyze the sensitivity of the performance with respect to quantization in **Table 9** in **Appendix C.3 Evaluation on the sensitivity of quantization level (line 1115)**. Specifically, the results indicate that BiHDTrans exhibits robustness with respect to the quantization level used in the HD value encoding. We believe this behavior is well-grounded because it aligns with: (1) prior HD computing studies (Rahimi et al., 2019); (2) prior studies on neural network quantization, since BiHDTrans can be viewed as a special form of neural network as it is trained as an equivalent BNN (Frankle & Carbin, 2019); (3) our theoretical analysis (Theorem 1), as BiHDTrans can still outperforms SOTA binary Transformer by 4.8\% when quantization level is reduced to 16.
>
> **Table 9: The sensitivity of quantization level of BiHDTrans.**
> | Quant. Level | JapaneseVowels | Heartbeat | SpokenArabicDigits | FaceDetection | PEMS-SF | RacketSports | Epilepsy | Average |
> |--------------|-----------------|-----------|----------------------|----------------|---------|---------------|----------|---------|
> | 256          | **97.30**       | **77.56** | **96.41**            | 61.18          | **87.28** | **83.82**     | **86.96** | **84.36** |
> | 128          | 96.76           | 75.61     | 96.36                | 60.61          | 86.82   | 83.47         | **86.96** | 84.36   |
> | 64           | 97.03           | 75.61     | 96.13                | **61.95**      | 84.39   | 82.66         | 86.23    | 83.80   |
> | 32           | 95.68           | 75.61     | 96.04                | 60.90          | 84.97   | 80.92         | **86.96** | 83.43   |
> | 16           | 96.49           | 75.12     | 95.59                | 59.76          | 82.66   | 81.58         | 86.23    | 82.49   |
>
> -	*Using a different font for (or italicizing) the accuracies of full-precision model in Table 1 is recommended, because the bolded results seem to be the best model performance excluding those of the full-precision model.*
>
> We confirm that the bolded results are the best model performance excluding the full-precision model. We have italicized the accuracies of full-precision model in **Table 1** and also the added **Table 10** as recommended.

---

> ### Author Response · Authors · 2025-11-22
> **Response to all questions -- Part 2**
>
> -	*Additional datasets of a larger scale are recommended to stress-test the model, such as PTB-XL and Sleep-EDF.*
>
> We would like to kindly expound that except of the InsectWingbeat dataset, where almost all SOTA models seems to failed to realize validate classification (Eldele et al., 2024), we have utilized 2 of 3 largest datasets in the UEA archive (SpokenArabicDigits, FaceDetection). However, we do acknowledge that some UEA archive datasets could be rather simple. Instead of simply adding larger classification datasets, we believe that extending our evaluation to anomaly detection task could be more meaningful, as it can not only stress-test the model, but also verify its generality. The results are provided in **Table 10** in **Appendix C.4 Extension to anomaly detection (line 1144)**, utilizing the NASA datasets SMAP and MSL. The results align with the MTSC results in Table 1, demonstrating that the proposed BiHDTrans generalizes effectively beyond classification and offers a compact, efficient solution for a more general time-series representation.
>
> **Table 10: Comparison to SOTA binary Transformers for anomaly detection.**
> | Model          | SMAP Precision | SMAP Recall | SMAP F0.5 Score | MSL Precision | MSL Recall | MSL F0.5 Score|
> |----------------|----------------|-------------|-----------------|---------------|-----------|----------------|
> | Full Precision | *0.592*        | *1.000*     | *0.645*         | *0.813*       | *0.929*   | *0.833*        |
> | BiBERT         | 0.511          | 0.691       | 0.539           | 0.719         | 0.885     | 0.747          |
> | BiT            | 0.513          | 0.357       | 0.471           | 0.710         | 0.846     | 0.733          |
> | BiViT          | 0.513          | 0.357       | 0.471           | 0.735         | **0.962** | 0.772          |
> | BiHDTrans      | **0.529**      | **0.945**   | **0.580**       | **0.774**     | 0.961     | **0.805**      |

---

### Official Review · Reviewer_Azyu · 2025-10-31

**Soundness:** 3
**Presentation:** 3
**Contribution:** 2
**Rating:** 2
**Confidence:** 3

**Summary:**

This paper proposes to introduce hyperdimensional (HD) computing to the transformer architecture for the task of multivariate time series classification (MSTC). The authors explain the pitfalls of transformer architecture, especially in MTSC, and argue that HD computing can solve them. They prove several theoretical results showing 1) that binarizing a neural network within the HD computing setup can be more beneficial than doing so in the vanilla transformer architecture, and 2) information distortion within their framework remains bounded.
The experimental results on a handful of UEA datasets show impressive computational gains and strong performance against other variations of compute-optimized transformers.

**Strengths:**

1.	An interesting exploration of applying HD computing to the transformer architecture with application to MTSC
2.	Several theoretical results showing why the proposed idea makes sense and should work reasonably in practice
3.	An experimental evaluation from both the accuracy and computational efficiency perspectives

**Weaknesses:**

1.	Lack of motivation for the MTSC task, specifically, where most models can run on edge devices. Although the authors do not mention it, SOTA TSFMs for classification (Mantis-8M (Feofanov et al. 2025), NuTime) are quite small and can be used easily on an edge device. It is not clear why we would want to use such involved techniques as HD computing in this setup.
2.	Evaluation is very limited, and no explanation for why only 7 datasets from the UEA repository were chosen is provided. There are now many MTSC datasets (see WOODS benchmark, for instance) that can be used to evaluate the proposed method as well.
3.	Outdated literature review omitting most of the recent advances in time series foundation models. As mentioned above, the authors do not mention the TSFMs proposed in the field in the last year or so, although they achieve strong performance in TSC.

**Questions:**

1. Can authors better motivate the need for using HD computing? For instance, one can take the MOMENT foundation model that is rather large and show the same findings with it? In this case, one would be able to see the usefulness of this approach.
2. Is there any reason for using only 7 UEA datasets?
3. Is there any particular reason to omit TSFMs from the introduction in Section 2? It seems that they are the most important use-case for the author's contribution, yet most references stop at 2023.

---

> ### Author Response · Authors · 2025-11-22
> **Response to all questions -- Part 1**
>
> Thank you for your comments. Below, we respond to all questions.
>
> -	*Can authors better motivate the need for using HD computing? For instance, one can take the MOMENT foundation model that is rather large and show the same findings with it? In this case, one would be able to see the usefulness of this approach.*
>
> We respectfully clarify that the reviewer’s concern regarding the motivation of BiHDTrans, as well as the attention to compare against TSFMs such as Mantis, stem from a misunderstanding of the research scope and level of our work.
>
> TSFMs like Mantis and NuTime represent **architecture-level or pretraining/data-processing innovations** . They optimize Transformer architectures (e.g., ViT) or pretraining strategies to achieve better performance on various time series tasks. For instance, Mantis employs a lightweight ViT-based architecture with a model size on the order of MBs, using standard Transformer modules as building blocks. While MB-level models can run on certain edge platforms, they still exceed the capacity of many micro-controllers (e.g., STM32, Arduino).
>
> In contrast, BiHDTrans is a **module-level contribution**. Our goal is to develop a binary, neuro-symbolic Transformer module (not a standalone architecture) to demonstrate its effectiveness through a one-block instantiation whose model size is in 10-KB level. We provide comprehensive comparisons against SOTA binary Transformer modules (BiBERT, BiT, BiViT), showing significant advantages in both accuracy and computational efficiency. Meanwhile, another benefit of this implementation is that we are able to consider BiHDTrans as an advancement in HD computing, thereby comparing with SOTA binary HD computing models and demonstrate the effectiveness of BiHDTrans from HD computing perspective.
>
> Importantly, BiHDTrans, as well as BiBERT, BiT, BiViT, can replace the standard Transformer blocks in architecture-level models like Mantis or NuTime to achieve full binary inference. While doing so inevitably involves trade-offs between efficiency and performance, what we are trying to stress here is the motivation and contribution of BiHDTrans remain valid and significant at the module level.
>
> Therefore, we believe comparing BiHDTrans directly with architecture-level TSFMs conflates two different research scales. Our motivation remains well-defined: to advance binary Transformer module via integrating HD computing principles and demonstrate its superiority over existing binary Transformer modules.

---

> ### Author Response · Authors · 2025-11-22
> **Response to all questions -- Part 2**
>
> -	*Is there any reason for using only 7 UEA datasets?*
>
> We selected these seven UEA datasets to cover a diverse set of IoT-related application domains, including physiological monitoring, spoken command recognition, GPS trajectory classification, and IMU-based activity recognition. In addition, our work includes not only software evaluation but also a full FPGA hardware implementation, which requires non-trivial engineering effort. For this reason, we did not exhaustively evaluate all UEA datasets or the hundreds of datasets occasionally used in TS classification studies.
>
> To further assess generality beyond MTSC, we extended our evaluation to anomaly detection. The results are provided in **Table 10** in **Appendix C.4 Extension to anomaly detection (line 1144)**, utilizing the NASA datasets SMAP and MSL. The results align with the MTSC results in Table 1, demonstrating that the proposed BiHDTrans generalizes effectively beyond classification and offers a compact, efficient solution for a more general time-series representation.
>
> **Table 10: Comparison to SOTA binary Transformers for anomaly detection.**
> | Model          | SMAP Precision | SMAP Recall | SMAP F0.5 Score | MSL Precision | MSL Recall | MSL F0.5 Score|
> |----------------|----------------|-------------|-----------------|---------------|-----------|----------------|
> | Full Precision | *0.592*        | *1.000*     | *0.645*         | *0.813*       | *0.929*   | *0.833*        |
> | BiBERT         | 0.511          | 0.691       | 0.539           | 0.719         | 0.885     | 0.747          |
> | BiT            | 0.513          | 0.357       | 0.471           | 0.710         | 0.846     | 0.733          |
> | BiViT          | 0.513          | 0.357       | 0.471           | 0.735         | **0.962** | 0.772          |
> | BiHDTrans      | **0.529**      | **0.945**   | **0.580**       | **0.774**     | 0.961     | **0.805**      |
>
> Finally, regarding the WOOD benchmark, we would like to clarify that this benchmark focuses on domain/covariate shift and other OOD generalization challenges, which are outside the intended scope of this work. Our paper focuses on developing an efficient binary Transformer module, and not on OOD generalization. Therefore, we do not include WOOD benchmark in this study.
>
> -	*Is there any particular reason to omit TSFMs from the introduction in Section 2?*
>
> Thank you for pointing out the omission of recent TSFMs. We acknowledge the importance of SOTA TSFMs, and have updated **Section 2.2 Time Series Foundation Models (line 98)** to include the latest TSFM works.

---

> > ### Comment · Reviewer_Azyu · 2025-11-25
> > **Thank you for the provided answers**
> >
> > Dear authors
> >
> > I want to thank you for providing the answers to my questions. I would still like to ask the authors to perform a more comprehensive evaluation of their approach either on the UEA, UCR, or WOODS collections (which can be used in IID setting and compared to the ERM baseline of the dedicated report). They are the widely used benchmarks allowing for a meaningful comparison of the proposed contributions. Unfortunately, without it, I would not be able to endorse this paper more strongly.

---

> ### Author Response · Authors · 2025-11-27
> **Response to Reviewer Azyu's follow-up**
>
> We thank the reviewer for the suggestion.
>
> We fully agree that a broad evaluation across large benchmark suites (e.g., UEA, UCR, WOODS) would be valuable. However, we believe our current evaluation already covers representative datasets across different domains (epilepsy EEG, spoken digits, ECG arrhythmia, wearable-sensor HAR), which demonstrate the core properties we highlight: (i) stable binarized-attention behavior, and (ii) substantial efficiency gains and improved effectiveness compared with SOTA binary Transformers and SOTA HD computing.
>
> Importantly, our contribution includes not only the software algorithm but also an algorithm–hardware co-design with a full FPGA pipeline. **Reproducing the requested benchmarks in a way that fairly measures both software and hardware trade-offs requires long experimental schedules and engineering effort that exceed the computational and time budget available during the rebuttal period.**
>
> We hope the reviewer finds the provided empirical evidence sufficient to assess the contributions within the intended scope of this work. Nevertheless, we fully respect the reviewer’s final judgment, and if there are any additional questions, we would be very happy to address them.

---

### Official Review · Reviewer_k5Rg · 2025-11-02

**Soundness:** 3
**Presentation:** 3
**Contribution:** 3
**Rating:** 8
**Confidence:** 4

**Summary:**

This paper proposes BiHDTrans, a neurosymbolic architecture that performs self-attention entirely in high-dimensional binary (hyperdimensional) space. Queries/keys/values are constructed via HD binding, enabling end-to-end binarization without softmax and heavy FFNs. The paper argues (via concentration-of-measure style analysis) that binarizing in holographic space incurs less distortion than directly binarizing neural parameters. Empirically, BiHDTrans improves MTS classification accuracy over SOTA binary Transformers while delivering very large latency gains on an FPGA pipeline and retains accuracy under sizable reductions in HD dimensionality.

**Strengths:**

• The unification of hyperdimensional computing and Transformer-style attention is conceptually elegant and technically novel, offering a neurosymbolic route to full binary inference.
• Hardware results substantiate the efficiency claim, with real FPGA measurements showing major latency and energy gains.
• Theoretical analysis, while asymptotic, provides an intuitive justification for why HD binarization preserves representational fidelity.

**Weaknesses:**

• Evaluation is limited to classification; forecasting or anomaly detection would strengthen generality.
• Finite-dimensional behavior is not explored; empirical distortion plots could help quantify theory–experiment alignment.
• Training details (learning rates, temperature parameters, or binarization schemes) are somewhat under-specified for reproducibility.

**Questions:**

Can the authors provide an empirical plot of distortion versus dimensionality to substantiate the concentration-based theory? How sensitive is model accuracy to the number of attention heads and HD dimension? Could a small forecasting or anomaly detection task be included to demonstrate broader applicability?

---

> ### Author Response · Authors · 2025-11-22
> **Response to all questions**
>
> Thank you for your comments. Below, we respond to all questions.
>
> -	*Can the authors provide an empirical plot of distortion versus dimensionality to substantiate the concentration-based theory?*
>
> We have added empirical plots of distortion versus dimensionality in **Figure 5** in **Appendix C.1 Empirical validation of Theorem 1 (line 1026)** and **Figure 6** in **Appendix C.2 Empirical validation of Theorem 2 (line 1039)** ,and also provided a more thorough analysis.
>
> -	*How sensitive is model accuracy to the number of attention heads and HD dimension?*
>
> As discussed in **Section 6.4 Exploring dimensionality tradeoffs**, we have already analyzed the sensitivity to HD dimension, which is the primary factor influencing representational capacity in hyperdimensional computing. Regarding the number of attention heads, we argue that this hyperparameter is generally not considered sensitive or critical in standard Transformer architectures. This choice is largely standardized in prior work, and mainstream Transformer models (e.g., ViTs, GPTs) rarely conduct sensitivity analyses on head count. Since the role and behavior of multi-head mechanism in BiHDTrans remain the same as in conventional Transformers, we believe that a dedicated sensitivity study on the number of attention heads is not essential for the scope of our contributions.
>
> However, we do provide an additional evaluation on the sensitivity of the quantization level in **Appendix C.3 (line 1113)**, which we believe is more valuable in terms our binary design. Specifically, the results in **Table 9 (line 1135)** indicate that BiHDTrans exhibits robustness with respect to the quantization level used in the HD value encoding. We believe this behavior is well-grounded because it aligns with: (1) prior HD computing studies (Rahimi et al., 2019); (2) prior studies on neural network quantization, since BiHDTrans can be viewed as a special form of neural network as it is trained as an equivalent BNN (Frankle & Carbin, 2019); (3) our theoretical analysis (Theorem 1), as BiHDTrans can still outperforms SOTA binary Transformer by 4.8\% when quantization level is reduced to 16.
>
> **Table 9: The sensitivity of quantization level of BiHDTrans.**
> | Quant. Level | JapaneseVowels | Heartbeat | SpokenArabicDigits | FaceDetection | PEMS-SF | RacketSports | Epilepsy | Average |
> |--------------|-----------------|-----------|----------------------|----------------|---------|---------------|----------|---------|
> | 256          | **97.30**       | **77.56** | **96.41**            | 61.18          | **87.28** | **83.82**     | **86.96** | **84.36** |
> | 128          | 96.76           | 75.61     | 96.36                | 60.61          | 86.82   | 83.47         | **86.96** | 84.36   |
> | 64           | 97.03           | 75.61     | 96.13                | **61.95**      | 84.39   | 82.66         | 86.23    | 83.80   |
> | 32           | 95.68           | 75.61     | 96.04                | 60.90          | 84.97   | 80.92         | **86.96** | 83.43   |
> | 16           | 96.49           | 75.12     | 95.59                | 59.76          | 82.66   | 81.58         | 86.23    | 82.49   |
>
>
> -	*Could a small forecasting or anomaly detection task be included to demonstrate broader applicability?*
>
> We have extended our evaluation to the anomaly detection task. The results are provided in **Table 10** in **Appendix C.4 Extension to anomaly detection (line 1144)**, utilizing the NASA datasets SMAP and MSL. The results align with the MTSC results in Table 1, demonstrating that the proposed BiHDTrans generalizes effectively beyond classification and offers a compact, efficient solution for a more general time-series representation.
>
> **Table 10: Comparison to SOTA binary Transformers for anomaly detection.**
> | Model          | SMAP Precision | SMAP Recall | SMAP F0.5 Score | MSL Precision | MSL Recall | MSL F0.5 Score|
> |----------------|----------------|-------------|-----------------|---------------|-----------|----------------|
> | Full Precision | *0.592*        | *1.000*     | *0.645*         | *0.813*       | *0.929*   | *0.833*        |
> | BiBERT         | 0.511          | 0.691       | 0.539           | 0.719         | 0.885     | 0.747          |
> | BiT            | 0.513          | 0.357       | 0.471           | 0.710         | 0.846     | 0.733          |
> | BiViT          | 0.513          | 0.357       | 0.471           | 0.735         | **0.962** | 0.772          |
> | BiHDTrans      | **0.529**      | **0.945**   | **0.580**       | **0.774**     | 0.961     | **0.805**      |
>
> -	*Training details (learning rates, temperature parameters, or binarization schemes) are somewhat under-specified for reproducibility.*
>
> Thank you for bring this to our attention, we have provided a more comprehensive training details in **Appendix A Hyperparameters of BiHDTrans  (line 760)** for reproducibility.

---

### Author Response · Authors · 2025-11-22
**Overall Response**

We sincerely thank all reviewers for their constructive feedback. We are encouraged that the reviewers found the proposed direction of integrating hyperdimensional computing with binary Transformers both interesting and promising. We have carefully addressed all questions, clarified technical points, and incorporated additional experiments and analyses as suggested. Below, we summarize the major concerns raised and our corresponding actions.

**1. Scope and motivation of BiHDTrans**

Several reviewers asked about the motivation for using HD computing and whether the proposed design can scale beyond MTSC tasks. We clarify that BiHDTrans is a module-level contribution aimed at advancing binary transformer module (block), rather than competing with architecture-level TSFMs. Its main goal is to demonstrate that a fully binary, neuro-symbolic attention mechanism is feasible, theoretically grounded, and hardware-efficient. This module can be plugged into larger architectures (e.g., ViT-style TSFMs), whereas TSFMs such as Mantis focus on architecture-level design. We further clarified this distinction in the rebuttal.

**2. Feasibility and stability of binary attention**

Reviewers raised concerns about the absence of softmax and the potential brittleness of binarized attention. We provided empirical sparsity measurements of the binarized attention score $B_a$ throughout training and validation (**Figure 3, Section 6.5**). The sparsity remains stable around 0.5, confirming the absence of pathological sparsity/density. We also reported the maximum output entropy (MOE) following prior work (BiBERT), further supporting stability.

**3. New experiments and analysis**

We substantially strengthened the paper with new empirical evaluations:

-	**Appendix C.1 & C.2**: Empirical validation of Theorem 1 and Theorem 2 with distortion–dimensionality plots (**Figures 5 and 6**).

-	**Section 6.5**: Evaluation of the binarized attention score $B_a$ (**Figure 3**), demonstrating stable and non-pathological behavior of hard binary attention during training and validation.

-	**Appendix C.3**: Detailed sensitivity study on quantization levels (**Table 9**), showing robustness from 256 down to 16 levels.

-	**Appendix C.4**: New evaluation on anomaly detection using NASA SMAP and MSL datasets (**Table 10**). Results are consistent with MTSC findings and demonstrate that BiHDTrans generalizes beyond classification.

These additions collectively strengthen the empirical grounding and demonstrate broader applicability.

**4. Regarding dataset complexity and selection**

MTSC was chosen as the initial domain due to its controlled scale, which is essential for hardware evaluation, and because it remains representative of edge AI applications. We also extended our evaluation to anomaly detection tasks to assess generality (**Table 10**). Tasks such as OOD robustness (e.g., WOOD benchmark) fall outside the intended scope of this work and relate more to architecture-level innovations rather than module-level contributions.

**5. Compatibility with Transformer features and LLMs**

We clarified that BiHDTrans is not inherently incompatible with positional encoding mechanisms like RoPE or ALiBi. These can be applied in the real-valued embedding space before value hypervectors are generated.  From a technical standpoint, extending our HD-Transformer architecture to LLM-scale models is entirely feasible.

Overall, we believe the changes and clarifications have substantially strengthened the paper. We thank the reviewers again for their thoughtful comments, which have helped us improve both the clarity and completeness of this work.

---

### Author Response · Authors · 2025-11-29
**Summary Rebuttal for New AC**

Dear AC,

We sincerely thank you for taking the time to evaluate our submission. Below we provide a concise summary of our work, the main concerns raised by the reviewers, and how we have addressed them.

**1. Key Contributions**

Our paper proposes **BiHDTrans**, a binary hyperdimensional Transformer module for multivariate time-series classification (MTSC), with the following core contributions:

-	**Neurosymbolic design:** We integrate hyperdimensional (HD) computing with Transformer-style attention to perform fully binary self-attention. Theoretical and empirical results show reduced information distortion compared to SOTA binary Transformers, and improved sequential modeling relative to SOTA HD computing models.
-	**Hardware efficiency:** A fully pipelined **FPGA implementation** demonstrates substantial efficiency gains over SOTA binary Transformers.
-	**Theoretical grounding and empirical validation:** We provide formal analysis on HD-space binarization and validate it across multiple datasets, including MTSC and anomaly detection.

**2. Addressing Reviewer Concerns**

During the rebuttal, we carefully responded to all reviewers’ questions and concerns:

-	**Scope, motivation and compatibility:** We clarified that BiHDTrans is a **module-level contribution**, not an architecture-level proposal. It demonstrates that a fully binary, neurosymbolic attention mechanism is feasible, theoretically grounded, and hardware-efficient. Furthermore, we clarified that it is compatible with positional encodings (RoPE, ALiBi) and can be integrated into larger architectures such as TSFMs or LLMs, providing a practical foundation for future extensions.
-	**Binary attention stability:** We measured the sparsity and maximum output entropy of the binarized attention, showing stable and non-pathological behavior (**Figure 3**).
-	**Experiments and broader applicability:** We extended evaluation to anomaly detection (**Table 10**), provided distortion–dimensionality plots validating the theoretical analysis (**Figures 5 & 6**), as well as quantization sensitivity analysis (**Table 9**).
-	**Dataset selection and feasibility:** While we focused on representative datasets across multiple domains, we explained why large benchmark suites (e.g., UEA, UCR, WOODS) are beyond the feasible scope during the rebuttal stage due to limit time budget and the workload for FPGA co-design.

**Summary**

We have addressed all reviewer concerns, strengthened the paper with additional experiments and analyses, and clarified the motivation, technical details, and generality of BiHDTrans. We hope the AC finds the paper’s contributions significant and the rebuttal responses satisfactory.

---

### Meta-Review · Area_Chair_9U6f · 2026-01-06

**Summary:**

This paper proposes BiHDTrans, a fully binary hyperdimensional Transformer module for efficient multivariate time series classification, with a strong emphasis on FPGA-based acceleration. Reviewers generally agree that the paper is technically ambitious and well executed: the integration of hyperdimensional computing with Transformer-style attention is interesting, the theoretical analysis provides useful intuition about binarization in HD space, and the hardware results convincingly demonstrate substantial latency and efficiency gains.

The main issue is not correctness or engineering quality, but rather insufficient clarity and consensus on the paper’s conceptual positioning, scope, and practical significance within the broader ICLR community. A key point of contention is the scope of contribution. While the authors frame BiHDTrans as a module-level advance rather than an architecture-level alternative to time-series foundation models (TSFMs), several reviewers found this distinction insufficiently compelling. In practice, the evaluation remains limited to relatively small-scale MTSC datasets, and it is unclear whether the proposed module addresses a pressing bottleneck that cannot already be handled by existing lightweight TSFMs or efficient Transformers on edge devices. As a result, the motivation for introducing substantial algorithmic and theoretical complexity in this setting remains debatable.

In addition, although the paper provides strong hardware-centric evidence, the machine learning contribution is perceived as narrower and more specialized, appealing primarily to a subset of the neurosymbolic or hardware-aware community. For ICLR, where broad conceptual insight and general methodological impact are key, the contribution appears somewhat niche, with limited demonstrated relevance to mainstream representation learning or foundation model research.

The rebuttal is thorough and addresses many reviewer questions by adding experiments (e.g., anomaly detection), clarifying theoretical claims, and improving presentation. Nevertheless, these additions mainly strengthen completeness rather than resolving the underlying concerns about scope, motivation, and generality. Given the lack of post-rebuttal reviewer convergence and the remaining disagreements, the paper ultimately falls below the acceptance threshold.

**Reviewer Concerns:**

The motivation for applying such a complex binary HD-Transformer to MTSC, given existing lightweight TSFMs and efficient baselines, remains insufficiently convincing.

Evaluation is limited in scale and task diversity, and does not clearly establish broad generality.

The contribution is primarily module- and hardware-centric, with limited conceptual impact for the wider ICLR audience.

Reviewer opinions remain polarized, with no clear convergence toward acceptance after rebuttal.

**Reviewer Scores:**

Based on the original reviews and the rebuttal, it is unlikely that the rebuttal would lead to sufficient upward score revisions to overcome the remaining concerns. Consequently, the overall evaluation remains below the acceptance threshold, supporting a rejection decision.

---

### Decision · Program_Chairs · 2026-01-26

Reject